

**Reconstructing seasonality through stable isotope and trace element analysis of the Proserpine**
**stalagmite, Han-sur-Lesse Cave, Belgium: indications for climate-driven changes during the last 400**
**years**
Stef Vansteenberge[1*], Niels de Winter[1], Matthias Sinnesael[1], Sophie Verheyden[2,1], Steven Goderis[1,3],
Stijn J. M. Van Malderen[3], Frank Vanhaecke[3] and Philippe Claeys[1].
[1]Department of Analytical, Environmental and Geochemistry, Vrije Universiteit Brussel, Pleinlaan 2, B-
1050 Brussels, Belgium.
[2]Royal Belgian Institute of Natural Sciences, Jennerstraat 13, B- 1000 Brussels, Belgium
[3]Department of Analytical Chemistry, Ghent University, Campus Sterre, Krijgslaan 281 S12, B-9000
Ghent, Belgium
**\*Corresponding author: Niels.de.winter@vub.be**



**Abstract**
Annually laminated speleothems allow the reconstruction of paleoclimate down to a seasonal scale. In
this study, an annually laminated stalagmite from the Han-sur-Lesse Cave (Belgium) is used to study
the expression of the seasonal cycle in northwestern Europe during the Little Ice Age. More specifically,
two historical 12-year-long growth periods (ca. 1593-1605 CE and 1635-1646 CE) and one modern
growth period (1960-2010 CE) are analysed on a sub-annual scale for their stable isotope ratios ($\delta^{13}$C
and $\delta^{18}$O) and trace element (Mg, Sr, Ba, Zn, Y, Pb, U) content. Seasonal variability in the proxies is
confirmed with frequency analysis. Zn, Y and Pb show distinct annual peaks in all three investigated
periods related to annual flushing of the soil during winter. A strong seasonal in phase relationship
between Mg, Sr and Ba in the modern growth period reflects a substantial influence of prior calcite
precipitation (PCP). In particular, PCP occurs during summers when recharge of the epikarst is low. This
is also evidenced by earlier observations of increased $\delta^{13}$C values during summer. In the 17th century
intervals, there is a distinct antiphase relationship between Mg, Sr and Ba, suggesting that varying
degrees of incongruent dissolution of dolomite control the observed seasonal variations. The
processes controlling seasonal variations in Mg, Sr and Ba in the speleothem appear to change
between the 17th century and 1960-2010 CE. The Zn, Y, Pb and U concentration profiles, stable isotope
ratios and morphology of the speleothem laminae all point towards increased seasonal amplitude in
cave hydrology and higher drip water discharge during the 17th century. These observations reflect an
increase in water excess above the cave and recharge of the epikarst, due to a combination of lower
summer temperatures and increased winter precipitation during the 17th century. This study indicates
that the transfer function controlling Mg, Sr and Ba seasonal variability varies over time. Which process
is dominant, either PCP or dolomite dissolution, is clearly climate-driven and can thus be used as a
paleoclimate proxy itself.
**Keywords:** Speleothem, seasonality, Little Ice Age, trace element concentrations, stable isotope ratios,
proxy transfer functions
**1. Introduction**
Speleothems have been successfully used to reconstruct paleoclimate on various time scales (Fairchild
and Baker, 2012), from tropical latitudes (e.g. Wang et al., 2001) to temperate areas (e.g. (Genty et al.,
2003). Their ability to hold distinct annual layering enables paleoclimate reconstructions down to
seasonal scale. The occurrence of visible annual laminae in speleothems has been reported from sites
all over the world (Baker et al., 2008). A common expression of this visible layering is an alternation of
dark compact laminae (DCL) and white porous laminae (WPL), as defined by Genty and Quinif (1996).
According to Baker et al. (2008), the origin of visible seasonal layering is related to seasonal variations



in drip rate, in drip water supersaturation and/or in cave climatology. However, in most cases, visible
seasonal layering is formed by changes in drip water discharge (Baker et al., 2008). Such changes in
drip rate often coincide with the presence of a varying degree of prior calcite precipitation (PCP). PCP
is the process of calcite precipitation upstream of the site of speleothem deposition (Fairchild et al.,
2000). An increase in PCP occurs when the ability of cave waters to degas increases. Therefore, a higher
degree of PCP is attributed to drier periods (Fairchild et al., 2000; Fairchild and Treble, 2009). Variations
in the amount of PCP have been observed on a seasonal scale (e.g. Johnson et al., 2006). The presence
of seasonally laminated speleothems in Belgian cave systems is known for several decades (e.g. Genty
and Quinif, 1996). The best known example is the Proserpine stalagmite collected from Han-sur-Lesse
Cave and first described by Verheyden et al. (2006). The speleothem has a well-expressed visual and
geochemical seasonal layering over the last 500 years according to layer counting and U/Th dating (Van
Rampelbergh et al., 2014). This geochemical layering is reflected by sub-annual variations of stable
isotope ratios ($\delta^{13}$C and $\delta^{18}$O). A thorough understanding of modern seasonal control on variations in
$\delta^{13}$C and $\delta^{18}$O in speleothem calcium carbonate results from rigorous monitoring of the conditions at
the sample site in Han-sur-Lesse cave as carried out by Van Rampelbergh et al. (2014) for the period

62 2012-2014.

In addition to the commonly used speleothem $\delta^{18}$O and $\delta^{13}$C proxies, the use of trace elemental
concentrations (e.g. Mg, Sr, Ba, Zn and U) as paleoclimate and paleoenvironmental proxies is becoming
standard practice in speleothem reconstructions (Fairchild et al., 2000; Regattieri et al., 2016). The use
of trace elements brings additional information that can be used to unravel seasonal variability in
speleothem chemistry. Examples of this include the use of trace element concentrations as proxies for
precipitation (Baldini et al., 2002; Warken et al., 2018), soil processes (Regattieri et al., 2016) or
changes in sediment supply (Regattieri et al., 2016) and can be used to identify volcanic ash fall events
from speleothem records (Jamieson et al., 2015).
The first objective of this study is to better characterize the geochemical layering by adding trace
element proxies to improve the understanding of processes driving the geochemical layering and to
further resolve the relation with seasonal climatic variability. In addition, this work also compares the
seasonal cycle within earlier identified cold periods (Verheyden et al., 2006; Van Rampelbergh et al.,
2015; Supp. Mat. Fig. 1) to present-day seasonal signals. To achieve this, two 12-year long stalagmite
growth periods (1593-1605 CE ± 30, hereafter P16 and 1635-1646 CE ± 30, hereafter P17) and a recent
analogue deposited between 1960-2010 CE (hereafter referred to as P19) are analysed on a sub-annual
scale for their stable isotopic ($\delta^{13}$C and $\delta^{18}$O) and trace elemental variations. This information is then
interpreted in terms of climatic changes during the last 400 years.





**2. Geological setting**

**2.1 Han-sur-Lesse Cave**

With a total length of approximately 10 km, the Han-sur-Lesse Cave system, located within a limestone belt of Middle Devonian age, is the largest known subterranean karst network in Belgium (Fig. 1A). The cave system was formed by a meander cut-off of the Lesse River within the Massif de Boine, which is part of an anticline structure consisting of Middle to Late Givetian reefal limestones (i.e. the Mont-d'Haurs and Fromelennes Formations (Fm.); Delvaux de Fenffe, 1985). The thickness of the epikarst zone above the cave is estimated to be around 40 m (Quinif, 1988). Studies have shown the local presence of dolomite in these Givetian limestones. Within the Mont-d'Haurs Fm., the biostromal limestones are alternated with fine-grained micritic limestones and dolomitic shales (Preat and Bultynck, 2006). Additionally, a recent study by Pas et al. (2016) on Middle Devonian outcrops has shown that dolomitized beds also occur within the limestones of the Fromelennes Fm.

The Han-sur-Lesse Cave is located ~200 km inland at an elevation of 200 m above sea level. The region is marked by a warm temperate, fully humid climate with cool summers, following the Köppen-Geiger classification (Kottek et al., 2006). In the period 1999-2013, annual temperatures averaged 10.2 °C and average annual rainfall amount was 820 mm yr$^{-1}$ in Rochefort, 10 km from Han-sur-Lesse (Royal Meteorological Institute). The study site is affected by a North Atlantic moisture source all year round (Gimeno et al., 2010) and the amount of precipitation does not follow a seasonal distribution. Calculations applying the Thornthwaite formula (Thornthwaite and Mater, 1957) show that there is a strong seasonal trend in the water excess, i.e. the amount of rainfall minus the amount lost by evapotranspiration, with water excess only occurring from October to April (Genty and Deflandre, 1998; Genty and Quinif, 1996).

The studied speleothem was retrieved from the Salle-du-Dôme, a 150 m wide and 60 m high chamber that formed by roof-collapse of the limestone (Fig 1B). The Salle-du-Dôme is well ventilated, as it is located close to the cave exit and connected through two passages to nearby chambers. Monitoring of cave atmosphere within the Salle-du-Dôme for the period 2012-2014 showed that in 2013 the temperature inside the chamber varied seasonally between 10.5 and 14.5 °C (Van Rampelbergh et al., 2014). Similar seasonal trends in temperature are observed for the drip water, but the average is 0.5 °C colder. The pCO$_2$ of the cave air averages around 500 ppmv for the whole year. Yet, in summer (July-August), a rapid and temporary increase to 1000 ppmv is observed. Also during summer, rainwater δ$^{18}$O and δD above the cave increase by 3 ‰ and 30 ‰ (VSMOW, Vienna Standard Mean Ocean Water) respectively, likely due to the atmospheric temperature effect as described by Rozanski et al. (1992). In contrast, drip water δ$^{18}$O and δD remain fairly stable throughout the year, with averages of -7.65 ‰





and -50.1 ‰ VSMOW and standard deviations of 0.07 ‰ and 0.6 ‰ VSMOW, respectively. During late
summer (September), an increase of 1.5 ‰ is observed in the $\delta^{13}$C record of dissolved inorganic carbon
(DIC) within the drip water
**2.2 Proserpine speleothem**
The Proserpine speleothem is a 2 m high, tabular shaped stalagmite. The speleothem has a surface
area of 1.77 m² and is fed by a drip flow with drip rates ranging between 100 and 300 mL min$^{-1}$. The
speleothem grew over a period of approximately 2 kyr and has thus an exceptionally high average
growth rate of 1 mm yr$^{-1}$. The large speleothem was drilled and a 2 m long core was retrieved. The
upper 50 cm of this core, dating back to approximately 1500 CE (Supp. Mat. Fig. 2), shows a well-
expressed layering of alternating DCL and WPL (Verheyden et al., 2006). Previous studies concluded
that multi-decadal simultaneous changes in different proxies (such as crystal fabric, growth rate, layer
thickness, and oxygen and carbon stable isotope ratios) indicate that these are controlled by common
climatic, environmental or anthropogenic factors, despite the observation that some parts of the
Proserpine speleothem appear to have been deposited out of isotopic equilibrium with the drip water
(Verheyden et al., 2006; Van Rampelbergh et al., 2015). Based on a detailed cave monitoring study at
the Proserpine site in the years 2012 to 2014, Van Rampelbergh et al. (2014) showed that $\delta^{18}$O and
$\delta^{13}$C of seasonally deposited calcite reflect isotopic equilibrium conditions and that variations of stable
isotope ratios are induced by seasonal changes. These seasonal changes in stable isotope ratios
correspond with the observed visible layering. The speleothem $\delta^{18}$O value is believed to reflect changes
in seasonal cave climatology. While drip water $\delta^{18}$O remains constant, calcite $\delta^{18}$O decreases with ~0.6
‰ in summer months, caused by temperature-dependent fractionation during calcite precipitation.
This fractionation was calculated to be -0.2 ‰ °C$^{-1}$. In contrast, $\delta^{13}$C reflects seasonal changes occurring
at the epikarst level. A ~1.5 ‰ increase of $\delta^{13}$C in drip water DIC during late summer is directly reflected
in the freshly deposited speleothem calcite. The enrichment in drip water $\delta^{13}$C values occurs shortly
after the observed decrease in drip water discharge, and therefore seasonal variations in the degree
of prior calcite precipitation in the epikarst has been hypothesized to be the main driver of seasonal
$\delta^{13}$C changes in the drip water (Van Rampelbergh et al., 2014).
**2.3 Dating**
The age-depth model of the Proserpine speleothem core has been established and discussed by Van
Rampelbergh et al. (2015) and is provided in the supplementary material (Supp. Mat. Fig. 2). This age-
depth model was constructed by using a combined approach of U-Th radiometric dating, based on 20
U-Th ages, and layer counting. This has shown that the amount of counted layers is in good agreement
with the U-Th ages (see Table 2 in Van Rampelbergh et al., 2015). However, at 9 to 10 cm from the top



of the core, a perturbation with heavily disturbed calcite occurs, making it impossible to construct a
continuous layer counted chronology. Remains of straw and soot were found within this perturbation,
suggesting that at that time, fires were lit on the speleothem's paleosurface (Verheyden et al., 2006).
Layer counting gave an age of 1857 ± 6 CE for the reestablishment of calcite deposition after the
perturbation and U-Th age-depth modeling showed that the start of the perturbation occurred at 1810
± 45 CE (Van Rampelbergh et al., 2015). Radiocarbon dating of the straw fragments embedded in the
calcite gave an age between 1760 and 1810 CE, with 95.4 % probability. The age of 1810 ± 45 CE is
used to restart the layer counting after the perturbation towards the bottom of the core. This gave an
age of 1593 to 1605 ± 30 CE for P16 and 1635 to 1646 ± 30 CE for P17. The more recent section P19
studied here is situated above the perturbation and its age could be confidently established through
annual layer counting at 1960-2010 CE.

## 3. Methods

### 3.1 Analytical procedures

The three growth periods studied are shown in Fig. 2 and their age is derived from an age-depth model
based on U-Th-dating and layer counting (Verheyden et al., 2006; Van Rampelbergh et al., 2015). For
$\delta^{13}C$ and $\delta^{18}O$ analysis, powder samples are acquired using a Merchantek Micromill
(Merchantek/Electro Scientific Industries Inc. (ESI), Portland (OR), USA, coupled to Leica GZ6, Leica
Microsystems GmbH, Wetzlar, Germany) equipped with tungsten carbide dental drills with a drill bit
diameter of 300 µm. The powders are stored in a 50 °C oven prior the analysis to avoid $\delta^{13}C$ and $\delta^{18}O$
isotopic contamination. Measurements for P16 and P17 are carried out on a Nu Perspective isotope
ratio mass spectrometer (IRMS) coupled to a Nucarb automated carbonate preparation device (Nu
Instruments, UK) at the Vrije Universiteit Brussel (Belgium). The $\delta^{13}C$ and $\delta^{18}O$ records of P16 and P17
consist of 201 and 116 data points, respectively, resulting in temporal resolutions of ~20 and ~10 data
points per year, respectively. The analysis of the P19 interval is an extension of the previously published
seasonally resolved 1976-1985 transect (Van Rampelbergh et al., 2014) and is carried out on a Delta
plus XL IRMS coupled to a Kiel III carbonate preparation unit (Thermo Fisher Scientific, Germany) also
at the Vrije Universiteit Brussel. For P19, a total of 350 samples are analysed, providing a temporal
resolution of ~7 data points per year. Within each batch of ten samples, the in-house reference
material MAR2-2, prepared from Marbella limestone and calibrated against NBS-19 (Friedman et al.,
1982) is measured together with the samples to correct for instrumental drift ($\delta^{13}C$: 3.41 ± 0.10 ‰ (2s)
VPDB; $\delta^{18}O$: 0.13 ± 0.20 ‰ (2s) VPDB. All results are displayed as ‰VPDB (Vienna Pee Dee Belemnite)
with the individual reproducibility reported as 2 standard deviation (SD) uncertainties. Averages of the





total 2 SD uncertainties for $\delta^{13}C$ and $\delta^{18}O$ are 0.03 ‰ and 0.09 ‰ for the Nu Perspective setup. With
the Delta plus XL setup these are slightly higher, being 0.04 ‰ and 0.10 ‰ for $\delta^{13}C$ and $\delta^{18}O$,
respectively (Van Rampelbergh et al., 2014).
Trace element variations are determined using inductively coupled plasma-mass spectrometry
complemented by a laser ablation sample introduction system (LA-ICP-MS) at Ghent University
(Belgium). The LA-ICP-MS setup consists of a 193 nm ArF*excimer Analyte G2 laser ablation system
(Teledyne Photon Machines, Bozeman, MT, USA) coupled to a single-collector sector field 'Element XR'
ICP-MS unit (Thermo Fisher Scientific, Bremen, Germany). The laser is used to sample adjacent
positions along a line segment parallel to the stalagmite's growth axis. The positions are ablated one-
by-one for 15 s with a laser spot size of 50 µm in diameter, a repetition rate of 30 Hz and a beam energy
density of 3.51 J cm$^{-2}$. The line segments for P16, P17 and P19 are drilled at 287, 249 and 445 individual
positions, respectively. Sampling via individual drilling points is preferred over the conventional
approach of continuous line scanning be-cause the single positions can be sampled longer, resulting in
an improved limit of detection. To carry out the analyses, the speleothem sections and reference
materials are mounted in a HELEX 2 double-volume ablation cell. The Helium carrier gas (0.5 L min$^{-1}$) is
mixed with Argon make-up gas (0.9 L min$^{-1}$) downstream of the ablation cell, and introduced into the
ICP-MS unit, operated in low mass-resolution mode. Transient signals for Magnesium (Mg), Aluminium
(Al), Silicon (Si), Phosphorus (P), Sulphur (S), Potassium (K), Iron (Fe), Manganese (Mn), Zinc (Zn),
Rubidium (Rb), Strontium (Sr), Yttrium (Y), Barium (Ba), Lead (Pb), Thorium (Th), and Uranium (U) are
monitored during analysis of the laser-induced aerosol. Cool plasma conditions (800 W RF power) are
used to reduce Argon-based interferences and to increase the sensitivity of the analysis. A gas blank
subtraction is performed on the data acquired at each position, based on the signal acquired 10 s prior
to the ablation. Precise and accurate trace element concentration data are obtained from offline
calibration, using seven international natural and synthetic glass and carbonate reference materials
BHVO-2G, BIR-1G, GSD-1G, GSE-1G, and MACS-3 (United States Geological Survey) as well as SRM 610
and 612 (National Institute of Standards and Technology). Ca is used as an internal standard, following
the assumption that the calcium carbonate in the speleothem is made up of 38 wt. % Ca. Based on the
reference materials and settings described, the repeatability for the produced elemental
concentration data is typically on the order of 5% relative standard deviation (RSD). Limits of detection
(LODs) are given in Table 1.
**3.2 Data Processing**
Frequency analysis is applied to study the variations in the different proxy signals, and allows
evaluating which of these proxies records the seasonal cycle. The ability of frequency analysis to assess



the potential of a proxy to record the seasonal cycle in speleothems and other incremental climate
archives is already recognized by Smith et al. (2009) and de Winter et al. (2017). Furthermore, the
method can identify multi-annual trends or variability at the sub-seasonal level. Frequency analysis is
performed using Fast Fourier Transformations (FFT) of the isotopic and trace element data in the
distance domain. The data are de-trended and padded with zeros. The power spectra are plotted as
simple periodograms with frequencies plotted in the distance domain (mm$^{-1}$) to allow intuitive
interpretation. The significance level (95%) is evaluated using Monte Carlo noise simulations. The
routine used operates in MATLAB® and is based on the scripts provided in Muller and MacDonald
(2000), which are explained in more detail in Bice et al. (2012).

An effective method to compare sub-annual variations of different proxies with each other is by
resampling multiple annual cycles at a regular interval and stacking the individual cycles (Treble et al.,
2003;Johnson et al., 2006; Borsato et al., 2007; de Winter et al. 2018). The advantage of this method
is that the phase-relations of the different proxies are preserved (Treble et al., 2003). Annual stacks
are created based on the moving averages to diminish the influence of low-frequency noise on the
annual stacks. The number of points used for moving averages is determined as a function of the
sampling resolution (i.e., 3-point moving average for stable isotope records and 5-point moving
average for trace element records, see Fig. 5). Proxy records with well-constrained seasonal variation
are used to define seasonal cycles. In this study, individual years are selected based on $\delta^{13}$C (minima)
for stable isotope records and Zn (maxima) for the trace element records. Stable isotope ratios and
trace element stacks are created separately (Fig. 2). For P16 and P17, all annual cycles are included in
the stack, except for the first and the last one, since there is no guarantee that these are entirely
represented in the record. For P19, only ten years were selected from the full record to avoid any effect
multi-decadal variability and to maintain an approach similar to that of P16 and P17. The years are
indicated by the red line in Fig. 5.
**4.   Results**
The concentration range of each proxy measured in the three different intervals is shown in Fig. 3. For
$\delta^{13}$C and $\delta^{18}$O, the average values and ranges (minima to maxima) in P19 are significantly higher than
those in P17 and P16. To illustrate the spread in the trace element records, the median is used instead
of the average as it is less sensitive to large concentration ranges and outliers. Al, Si, K, Mn, Rb and Th
are not included in this study since > 25% of the data falls below the LOD. An exception is made in the
case of Y; few data points are retained for P17 (81% of the data is < LOD in P17 and 18% and 36% of
the data is < LOD in P19 and P16 respectively. However, Y data are discussed because of the clear
seasonal signal shown in P19 and P16 (Supp. Mat. Fig. 3 and 5).





Records of stable isotope ratios ($\delta^{13}$C and $\delta^{18}$O) and trace element (Mg, Zn, Sr, Y, Ba, Pb and U)
concentrations are plotted in the distance domain in Fig. 5. The occurrence of darker laminae (DCL) in
the samples is indicated by blue bands, clearly showing that layers are thicker in P16 and P17 (average
1.135 mm and 1.096 mm, respectively) compared to P19 (average 0.382 mm). For all intervals, the
seasonal cycles are well constrained by $\delta^{13}$C, with lower $\delta^{13}$C values occurring in DCL. The average $\delta^{13}$C
is higher for P19 (-8.36 ‰) compared to P17 and P16 (-9.82 ‰ and -10.04 ‰, respectively). In addition,
the amplitude of the individual cycles is larger in P19. Seasonal cycles in $\delta^{18}$O are much less
pronounced. The most distinctive cycles are observed in P19 and some can be identified in parts of
P17 and P16 (e.g. between 4 and 7 mm in P16 or between 3 and 7 mm in P17), while for other parts
(e.g. between 7 and 11 mm in P16) they appear to be absent.
Seasonal variations are observed for Mg, Sr and Ba in all three intervals investigated (Fig 5). In P17 and
P16, the median concentrations of these elements are closely related; 447 and 444 µg g$^{-1}$ for Mg, 51
and 45 µg g$^{-1}$ for Sr and 36 and 33 µg g$^{-1}$ for Ba (Fig. 3). However, in P19 concentrations of Mg and Ba
are slightly higher compared to the older intervals, i.e. 706 µg g$^{-1}$ for Mg and 46 µg g$^{-1}$ for Ba. This is
also the case for Pb and U with concentrations significantly lower in P17 (0.14 and 0.05 µg g$^{-1}$,
respectively) and P16 (0.14 and 0.07 µg g$^{-1}$, respectively) and a seasonal cycle that is less pronounced
than in P19 (0.37 and 0.18 µg g$^{-1}$). In contrast, P16 has the highest median concentrations of Zn (54 µg
g$^{-1}$) and Y (0.04 µg g$^{-1}$) and both elements display a well-defined seasonal covariation. Although the
concentration of Zn is lower in P19 and P17 (14 and 25 µg g$^{-1}$, respectively), the seasonal cycle is still
present. Similar observations can be made for Y in P19 (0.02 µg g$^{-1}$). Within P16 and P17, maxima of
Zn, Y, Sr and Ba mostly occur within the DCL.
Figure 4 shows an example of the FFT periodograms of $\delta^{13}$C, Mg, Zn and P in P16. Additional
periodograms for the other elements in P16, P17 and P19 are included as supplementary data (Supp.
Mat. Fig. 3-5). The frequency analysis confirms the clear seasonal cyclicity of $\delta^{13}$C previously observed
by Van Rampelbergh et al. (2014) (Fig. 4). The dominant frequency of $\delta^{13}$C in P16 is 0.8 mm$^{-1}$ (Fig. 4).
This corresponds to a period of 1.25 mm, which is in good agreement with an observed average layer
thickness of 1.13 mm (Supp. Mat. Fig. 6). Because of its distinct seasonal cyclicity, the $\delta^{13}$C cycle is used
as a reference to deduce whether or not other proxies record the seasonal cycle. Mg and Zn appear to
track this seasonal cycle well as their periodograms contain peaks at 0.8 and 0.75 mm$^{-1}$ respectively,
corresponding closely to the frequency of $\delta^{13}$C. For Zn, a broader double peak is observed with a main
period of 1.18 mm and a smaller period of 1.02 mm. This double peak in the periodogram is caused by
small variations in the thickness of the annual cycles around an average thickness of 1.14 m with a
lightly skewed distribution towards thinner layers (see Supp. Mat. Fig. 6). The P record doesn't display



any significant seasonal cycle (95% confidence) (Fig. 4). For P19, visible layers are thinner (average
0.382 mm) and also the variation in thickness is larger (RSD 28.9%) compared to P16 and P17 (Supp.
Mat. Fig. 6). This results in broader and less well defined seasonal peaks in the periodograms.

## 5.    Discussion

### 5.1 Seasonal cyclicity in trace element records

A schematic overview of the observed changes in the proxies discussed below and the interpretation
for the three intervals is provided in Table 2. Assessing the exact phasing of the seasonal cycles of
different trace elements to $\delta^{13}C$ and the visible layering remains challenging since 1) often a multitude
of factors control trace element variations within speleothems and 2) stable isotope ratios and the
trace element concentrations are not measured on the same samples. An example of such a phase
problem is the occurrence of an additional year in P16 in the trace element curve compared to $\delta^{13}C$
(Fig. 5, between 1 and 6 mm). Nevertheless, overall, $\delta^{13}C$ minima occur in the DCL, suggesting a similar
timing (and maybe control) on the visible laminae and $\delta^{13}C$ cycles. Trace element proxies show cyclicity
with a similar frequency as the $\delta^{13}C$ (Fig. 4), in contrast to $\delta^{18}O$ which seasonal cycles in P16 and P17
are less clear.

### 5.1.1 Zinc, yttrium and lead proxies

In earlier monitoring studies carried out in the Père-Noël Cave (also part of Han-sur-Less Cave system,
Fig. 1), the presence of a late autumn increase in discharge was identified (Genty and Deflandre, 1998;
Verheyden et al., 2008). In-situ conductivity measurements indicated an elevated mineral and/or
organic matter increase during this autumnal increase in drip water discharge (Genty and Deflandre,
1998; Verheyden et al., 2008). Measurements of the drip water discharge above the Proserpine
stalagmite show that in late November, a doubling of the discharge volume occurs. This increased
discharge is maintained until May, when a gradual decrease is observed (Van Rampelbergh et al.,
2014). The timing of the elevated discharge agrees with the theoretical water excess occurring above
the cave (Genty and Quinif, 1996). The observed seasonal cycle in Zn, Y and Pb in the intervals studied
is likely caused by this annual winter flushing. Variations in these trace metal concentrations within
annual speleothem layers have previously been attributed to the annual hydrological cycle. For
instance, Borsato et al. (2007) linked the peak concentrations of F, P, Cu, Zn, Br, Y and Pb to the annual
increase of soil infiltration during autumnal rainfall. Furthermore, it was suggested that the transport
of such elements mainly occurs via natural organic matter (NOM) or other form of colloidal material.
Enrichments of these soil-derived elements within speleothems are believed to be associated with high
drip water flows (Fairchild and Treble, 2009). Studies have shown that trace metals, such as Cu, Ni, Zn,



Pb, Y and REE, are predominantly transported via complexing by NOM, of which the fraction size in the
karstic waters ranges from nominally-dissolved to colloidal-to-particulate (Hartland et al., 2012; Wynn
et al., 2014). In the case of Zn and Pb, Fairchild et al. (2010) have shown that in Obir Cave (Austria) the
visible and ultra-violet lamination forms during autumn and is enriched in Zn, Pb and P. According to
Wynn et al. (2014), the correspondence of distinct Zn and Pb peaks with the autumnal laminae is
compelling evidence for a high-flux transport of these trace metals with NOM. However, in this study
no distinct annual cycle within the P record is observed (Fig. 4). Phosphorus is considered soil derived
as it originates from vegetation dieback (e.g. Baldini et al., 2002). Therefore, P has shown similar
variations as observed in Zn, Y and Pb in previous studies (Borsato et al., 2007; Fairchild et al., 2010).
In the Proserpine speleothem, no relation between P and other soil derived trace elements is detected.
An explanation for this can be similar to that proposed by Frisia et al. (2012), being that P is not derived
from soil leaching, but from other sources such as phosphate minerals present in the epikarst or
microbiological activity.
Because of the distinct signature of the seasonal cycle in Zn, the Zn peaks are used as tie-points to
create the annual stacks of other trace element records (Fig. 6), with lower concentrations occurring
during periods of lower discharge and vice versa. The much higher Zn and Y peaks in P16 compared to
P17 and P19 suggest an increased seasonality effect in discharge; therefore the accompanied annual
flushing of the soil above cave appears more intense in the early 17[th] century. Concentrations of Pb
are significantly higher in P19 compared to the other periods (median of 0.37 $\mu g\ g^{-1}$ versus 0.14 $\mu g\ g^{-1}$
and 0.14 $\mu g\ g^{-1}$ in P16 and P17, respectively). An increase of Zn and Y in P19 similar to that in Pb is not
observed, suggesting that the Pb-enrichment occurs at the soil level from another source. A study of
Allan et al. (2015) on Pb isotope ratios in the in the same Proserpine stalagmite shows that the Pb
concentrations are soil derived and originate from various sources of anthropogenic atmospheric
pollution (coal, industrial activities, steel production and road dust). This explains well the observed
higher Pb concentration in P19. Allan et al. (2015) identified increases in Pb concentration during 1945-
1965 CE and 1975-1990 CE, which are in agreement with the observed higher Pb concentrations in this
study between 20-18 mm and 13-5 mm. They also concluded that this 20[th] century anthropogenic
pollution only affects Pb and none of the other elements used as paleoseasonality proxy in this study.
**5.1.2 Magnesium, strontium and barium proxies**
Figure 6 shows that the annual stacks of Sr and Ba exhibit correlate strongly within all three intervals,
evidenced by Pearson correlation coefficients (*r*) of 0.71, 0.97 and 0.82 for P19, P17 and P16,
respectively with p-values much smaller than 0.01 (99% confidence level). Magnesium displays an
antiphase relationship with Sr and Ba in P16 (*r* = -0.85, p-value = $1.8*10^{-7}$) whereas in P19 this



relationship is in phase ($r$ = 0.64, p-value = 7.2*10$^{-4}$). For P17, there is no significant relationship
between Mg with Sr and Ba (low correlation, $r$ = -0.13, p-value = 0.53). A strong covariation of Mg with
Sr and Ba, as observed in P19, has previously been attributed to reflect the presence of prior calcite
precipitation (PCP) in the epikarst above the cave, caused by the occurrence of drier periods (Fairchild
et al., 2000), even on a seasonal scale (Johnson et al., 2006). The presence of PCP during late summer
(with high evapotranspiration above the cave), when strongly reduced drip water discharge exists
above the Proserpine stalagmite, has also been evoked to explain the enriched $\delta^{13}$C of freshly
deposited calcite during the cave's summer mode (Van Rampelbergh et al., 2014). Despite the
difficulties of accurately correlating trace elements and stable isotope proxies, there appears to be a
good agreement between the P19 Mg and $\delta^{13}$C record, with maxima in Mg corresponding with maxima
in $\delta^{13}$C, confirming the hypothesis of PCP control on these proxies. In contrast, the antiphase variation
in Mg with respect to Sr and Ba observed in P16, suggests the involvement of other processes that
dominate over PCP. A positive relationship between the Mg partition coefficient and temperature
would be expected from thermodynamic considerations, and this has indeed been observed in
experimental carbonate precipitation studies (Gascoyne, 1983; Rimstidt et al., 1998; Huang and
Fairchild, 2001; Day and Henderson, 2013). In similar experiments, strontium partitioning into
inorganic carbonate is known to remain constant with increasing temperatures but can be influenced
by calcite precipitation rate (Day and Henderson, 2013). Faster precipitation of calcite causes an
increased amount of lattice defects, resulting in an increased value for the partition coefficient of Sr
(Pingitore and Eastman, 1986) and thus more Sr uptake in the calcite. Higher temperatures, combined
with a decrease in drip water discharge, leading to decreased growth rates, could therefore
theoretically explain the antiphase relationship of Mg and Sr. However, growth rates in P16 are rather
high and additionally, it has been suggested that the variations of Sr and Mg in drip water chemistry
are often significantly higher than those caused by the processes mentioned above. Roberts et al.
(1998) concluded that the temperature-dependence of the Mg partition coefficient could theoretically
explain the observed seasonal Mg variations, but not the multi-annual trends, for which hydrological
changes are likely more important. Such observations have caused the interpretation of the Mg proxy
to shift from a temperature relationship to an interpretation in terms of hydrological chances such as
amount of water recharge in the epikarst (Fairchild and Treble, 2009). In this study, a more likely
explanation for the P16 antiphase relation in Mg, Sr and Ba is the incongruent dissolution of dolomite
(CaMg(CO$_3$)$_2$; IDD), taking place during annual periods that are characterized by enhanced water-rock
interaction. The presence of dolomite within lateral-equivalent Givetian limestone deposits in Belgium
has been recognised by Pas et al. (2016). Dolomitized parts of the limestone host-rock were observed
within the nearby Père-Noël Cave (Fairchild et al., 2001). During periods of decreased recharge, i.e.
summer for the Han-sur-Lesse Cave, prolonged interaction between water and rock leads to saturation



of the karstic water with respect to CaCO$_3$. When saturation is reached, incongruent dissolution of
dolomite (IDD) will start and Ca$^{2+}$ concentration remains constant due to the precipitation of calcite
(Lohmann, 1988). IDD increases the Mg/Ca of the drip water (Fairchild et al., 2000), but lowers the
Sr/Ca and the Ba/Ca, because dolomite tends to have lower Sr and Ba contents with respect to calcite
(Roberts et al., 1998). The IDD process is believed to overwhelm the PCP signal in P16 and is likely
responsible for the observed antiphase relation. During winter recharge, saturation of the water in the
epikarst with respect to calcite is not attained and dolomite does not dissolve.
The comparison of the annual stacks for Mg, Sr and Ba of the different intervals corroborates the idea
that PCP is the main process controlling the seasonal variations of these trace elements in P19 based
on the in-phase relation of Mg, Sr and Ba. Within P16, enhanced seasonality in recharge causes IDD to
dominate over PCP. This explains the antiphase relation of Mg against Ba and Sr. Somewhere between
the P16 and P19 periods, a turnover in the hydrological regime of the epikarst allowed PCP to become
dominant over IDD in the seasonal variations in the proxies. Within P17, the relationship between Mg,
Sr and Ba is less clear. This could point towards a change in hydrological regime between the periods
of deposition of P16 and P19.

**5.1.3 Uranium**

In speleothems, U is thought to be mainly derived from bedrock dissolution (Bourdin et al., 2011;
Jamieson et al., 2016) and to be subsequently transported by the ground water towards the
speleothem (Fairchild and Baker, 2012). The partition coefficient of U is <1 for calcite (Johnson et al.,
2006; Jamieson et al., 2016). This causes U to be preferentially excluded from the calcite and enriched
in the remaining drip water during the process of PCP. However, in P19, where PCP is evoked as the
dominant process controlling Mg, Sr and Ba seasonal variations, an antiphase relationship of U with
Mg, Sr and Ba is observed (Fig. 6). Johnson et al. (2006) concluded that scavenging of U as uranyl ion
(UO$_2$$^{2+}$) from the drip water onto the calcite crystal surfaces during PCP has a more dominant control
on seasonal U variability than the partition coefficient. Such mechanism explains why U is antiphase
with the Mg, Sr and Ba variations.

**5.2 Seasonal variations in δ$^{13}$C and δ$^{18}$O**

To compare and understand the seasonal variations in δ$^{13}$C and δ$^{18}$O, annual stacks were created (Fig.
7) by virtual resampling based on the occurrence of peaks in δ$^{13}$C values as this proxy reflects the
seasonal cycle best (Fig. 4). The minima in δ$^{13}$C always occur in DCL, for P16 and P17. In P19, this
relation is less clear, however on close inspection nearly all of the δ$^{13}$C minima occur within the DCL
(Fig. 5). Van Rampelbergh et al. (2014) suggested that seasonal changes in δ$^{13}$C of recent calcite are



driven by changes in PCP. Higher $\delta^{13}$C values occur when more PCP is observed, i.e. during periods of
lower recharge. The in phase variations of Mg, Sr and Ba in P19 described above supports the
hypothesis of a seasonally changing degree of PCP. Seasonal variations in the amount of PCP and its
effect on $\delta^{13}$C has previously been recognized in monsoon regions (Johnson et al., 2006; Ridley et al.,
2015). During P16, seasonal changes in incongruent dolomite dissolution dominate the trace element
variations of Mg versus Sr and Ba over PCP. However, since the main source of carbon in Han-sur-Lesse
cave waters is the vegetation cover above the cave (Genty et al., 2001), IDD is not expected to change
the $\delta^{13}$C signal. For example, a case study carried out by Oster et al. (2014) showed that an increase in
IDD did not affect the $\delta^{13}$C of the speleothem significantly, despite a difference of ~0.5 ‰ in $\delta^{13}$C
between the limestone and dolomite component in the host rock. Since $\delta^{13}$C is not affected by IDD,
the influence of PCP on the $\delta^{13}$C remains observable. Indeed, similar as in P19, for both P17 and P16
$\delta^{13}$C minima occur within DCL, suggesting that these DCL layers were deposited by during seasonal
periods of increased drip water discharge.
Observations from cave monitoring have shown that seasonal changes in cave temperature (11°C -
15°C) are the main driver of $\delta^{18}$O variations in freshly deposited calcite (-7.0‰ - -6.2‰; Van
Rampelbergh et al., 2014). The $\delta^{18}$O periodograms show that the seasonal cycle is less developed
compared to $\delta^{13}$C (Fig. 4 and Supp. Mat. Fig. 3-5). This is also expressed in the annual stacks (Fig. 7).
For P19, there is tendency towards a positive correlation of $\delta^{13}$C and $\delta^{18}$O but in P17 and P16 this is
unclear. Although analysis of recent calcite have clearly shown that $\delta^{18}$O values are at least partly
controlled by the cave temperature, interpretation of the seasonal $\delta^{18}$O changes is difficult due to the
reduced seasonal cyclicity in the $\delta^{18}$O records compared to other proxies. However, average $\delta^{18}$O
values of speleothem calcite are obviously more depleted for P17 and P16 compared to P19 (Fig. 3 and
Fig. 5). The hypothesis put forward here is that the lower average $\delta^{18}$O values of P16 point towards an
increase in winter precipitation above the cave, since Van Rampelbergh et al. (2014) has shown that
winter precipitation, such as the presence of snow, above Han-sur-Lesse cave causes a severe decrease
in $\delta^{18}$O of the precipitation. Subsequently, this decrease is then transferred to the drip water and into
the speleothem calcite.
**5.3 Variability in the seasonal cycle**
The observed changes of the seasonal variations in Mg, Sr and Ba between P19, P17 and P16 can only
be explained by a change in the process controlling the seasonal variability in Mg, Sr and Ba. In the
recent period, between 1960 and 2010 CE, PCP is identified as the main driver for seasonal changes in
Mg, Sr, Ba trace element concentrations. This hypothesis is supported by the $\delta^{13}$C variations. In the
17[th] century intervals, Mg, Sr and Ba variations suggest that incongruent dissolution of dolomite rather



than PCP, dominates the seasonal signal. Fairchild and Baker (2012) defined the term transfer function
to describe the quantitative relation between the speleothem chemistry and changing cave
environments or climate. In this case, there is a change in (qualitative) transfer function from
incongruent dolomite dissolution to prior calcite precipitation. This change in transfer function is likely
to be climate-controlled since there are no indications for drastic changes in cave morphology over the
last 500 years, as interpreted from the long term stable isotope ratio record (Van Rampelbergh et al.,
2015 and Supp. Mat. Fig. 1). It is known that the strength of the acting transfer function can be used
as a paleoclimate proxy. For example, Jamieson et al. (2016) demonstrated that the seasonal (anti-
)correlation between $\delta^{13}$C and U/Ca varies through time within a Common Era stalagmite from Belize.
During drier years, reduced seasonal variability in prior aragonite precipitation causes U/Ca and $\delta^{13}$C
to correlate more positively compared to wetter years. This illustrates how a transfer function can be
regarded as a valuable paleoclimate proxy. In any case, a certain threshold must be reached for a
switch between transfer functions to take place. A prerequisite for PCP to occur is the presence of
sufficient karstic voids filled with a gas phase characterized by a lower $pCO_2$ than that with which the
infiltrating waters previously equilibrated (Fairchild and Treble, 2009). The presence of such karstic
voids is dependent on the multi-annual to decadal recharge amount of the karstic aquifer. Indeed, the
average values of trace element concentrations imply an increased water availability during P16 and
P17 compared to P19. More specifically peaks in soil-derived trace element concentrations (Zn and Y)
are higher for P16, pointing towards enhanced flushing and an increased seasonality in water
availability. An anthropogenic influence explains the higher concentrations of Pb in P19. In addition,
trace element concentrations originating from host rock dissolution (Mg, Sr, Ba and U) are significantly
lower for P16, resulting from lower multi-annual water residence time. Lastly, layers in P16 and P17
are up to three times thicker compared to P19 (Fig. 2 and Supp. Mat. Fig. 6), which reflects higher
growth rates. The positive relationship between water supply and growth rate has been demonstrated
in the past (Baker et al., 1998; Genty and Quinif, 1996). In large and irregular shaped stalagmites, such
as the Proserpine, within-layer thickness can often be quite large (Baker et al., 2008). The long-term
layer thickness evolution shows a clear difference between the 17[th] century and present day. The
significantly thinner layers during recent times clearly indicate that less water is available compared to
the 17[th] century.
A straightforward explanation for the observed wetter cave conditions during, in particular, P16 is an
increase in seasonal water excess. An elevated water excess can be caused by an increase in
precipitation or a decrease in temperature. A lower temperature, especially during summer, results in
a decreased evaporation of surface water. Calculations of present-day potential evapotranspiration
(PET) with the Thornthwaite equation (Thornthwaite and Mather, 1957) for the period 1999-2012



show a negative water excess lasting from May to September (Fig. 8). Although the Thornthwaite and
Mather (1957) method does not include vegetation effects, it is still a reliable tool to provide an idea
of the effect of changes in the temperature and/or precipitation on the PET (Black, 2007). The effect
of a temperature decrease during summer months on the water excess was simulated with an
arbitrarily chosen 1°C temperature drop compared to the 1999-2012 average monthly temperature.
Such a temperature drop appears to have only a minor influence (Fig. 8). A hypothetical increase of
total annual rainfall with 200 mm, equally spread across 12 months, has a much larger effect on the
water excess (Fig. 8). However, this would decrease the length of the annual interval during which no
recharge occurs (i.e. only during June-July instead of May-September) providing less suitable
conditions for dolomite dissolution to occur. Therefore, the most plausible explanation would be to
have a stronger seasonal distribution in the amount precipitation (with more winter precipitation),
whereas today no seasonality in the amount of rainfall is observed.
**5.4 Implications for 17th century paleoclimate**
The majority of Common Era paleoclimate reconstructions are based on tree-ring data (D'Arrigo et al.,
2006), although other records, for example historical documents (e.g., Dobrovolny et al., 2010), ice
cores (e.g., (Zennaro et al., 2014) or speleothems (e.g., Baker et al., 2011; Cui et al., 2012) are used as
well. Over the last decades, consensus has been reached that changes in solar irradiance and volcanic
activity are the main drivers of short-term natural climate variability during the last millennium (e.g.
Crowley, 2000; Bauer et al., 2003). Interpretations of the stable isotope and trace element proxies
obtained on the Proserpine speleothem show that a higher recharge state of the karstic aquifer
characterizes the 17th century intervals compared to 1960-2010. Such an increase in recharge requires
a decrease in evapotranspiration, which can result either from lower summer temperatures or higher
total annual precipitation. Although it is difficult to discriminate between both, the effect of a total
annual precipitation increase on the recharge is expected to be higher compared to a decrease in
summer temperature (Fig. 8). Globally dispersed regional temperature reconstructions indicate that
multi-decadal warm or cold intervals, such as the Medieval Warm Period or the Little Ice Age (LIA), are
not global events. Yet, a global cooling trend starting at 1580 CE is observed in the majority of the
reconstructions (PAGES 2k Consortium, 2013). Several paleoclimate reconstructions agreed upon the
occurrence of a cold period around 1600 CE, with negative temperature anomalies persisting in Europe
at decadal and multi-decadal scales (Ljungqvist et al., 2012; Luterbacher et al., 2016; Masson-Delmotte
et al., 2013). Reconstructions of European summer temperature provided by Luterbacher et al. (2016)
indicate that the coldest 11 and 51 year period since 755 CE in the area of Han-sur-Lesse cave occurred
within the 17th century. These reconstructions showed a summer temperature decrease of 1 – 1.5°C
around 1600-1650 CE. Although the 17th century has been recognized as the coldest of the past twelve



centuries, hydrological climate conditions appear close to the long-term mean (Ljungqvist et al., 2016),
with no significant wetting or drying trend. However, to account for the differences between the 1960-
2010 interval and the 17th century observed in this study, an increase in the amount winter
precipitation is needed, suggesting that climatic conditions were wetter during that time. Such a
hypothesis is also supported by the depleted $\delta^{18}O$ values in P16, indicating an increase in winter
precipitation.
**6. Conclusions**
This study of annual trace element and stable isotope ($\delta^{13}C$ and $\delta^{18}O$) variations over three different
time intervals of the annually laminated Proserpine stalagmite from the Han-sur-Lesse Cave (Belgium)
shows that seasonal changes in Mg, Sr and Ba during the recent period (1960-2010) suggest a strong
effect of prior calcite precipitation, caused by lower water availability during summer. In the 17th
century (1600 CE ± 30 and 1640 CE ± 30), however, Mg is in antiphase with Sr and Ba. This implies that
another process overwrites the PCP dominated seasonal cycle in these trace elements. A varying
degree of incongruent dolomite dissolution is the most plausible hypothesis, with more dissolution
occurring during summer when water residence times in the epikarst are longer. The transfer function
governing the trace elements, PCP or a varying degree of dolomite dissolution, depends on water-rock
interaction. Stable isotope ratios ($\delta^{13}C$ and $\delta^{18}O$), soil derived trace element concentrations (Zn, Y and
Pb) and speleothem morphology indicate that the multi-annual recharge of the epikarst was higher in
the 17th century. The change in the response of  Mg, Sr and Ba in the Proserpine speleothem to
environmental changes was identified to be climate-driven and likely results from a recharge increase
caused by a combination of lower summer temperatures and an increase in the amount of winter
precipitation in the 17th century for the Han-sur-Lesse cave region. The effect of an increase in winter
annual precipitation on the recharge is expected to be larger compared to a decrease in summer
temperature. The data obtained in this study clearly shows a stronger seasonal cycle in cave hydrology
during the 17th century.
This high-resolution, multi proxy study provides a good example of how seasonal proxy transfer
functions of trace elements in speleothem calcite can change over time. Such an observation has
implications for future speleothem-based paleoclimate reconstructions, since transfer functions for
specific cave sites, determined by cave monitoring, are often assumed to remain constant when no
drastic changes in the cave environment have occurred. As the change in trace element proxy transfer
function observed in this study is climate-driven, this change by itself can serve as a valuable
paleoclimate proxy.



**Author contributions**

Stef Vansteenberge and Sophie Verheyden designed the study. Stef Vansteenberge, Steven Goderis and Stijn Van Malderen carried out LA-ICP-MS measurements. Stef Vansteenberge, Matthias Sinnesael and Niels de Winter carried out stable isotope measurements. Stef Vansteenberge carried out the data processing and plotting with contributions from Steven Goderis, Niels de Winter and Matthias Sinnesael. Frank Vanhaecke and Philippe Claeys provided laboratory facilities and supported the measurements. Stef Vansteenberge, Niels de Winter and Matthias Sinnesael prepared the manuscript with contributions from all co-authors.

**Acknowledgements**

All authors thank the Domaine des Grottes de Han S.A. for allowing us to sample the stalagmites and carry out other fieldwork. Special thanks for M. Van Rampelbergh, whose PhD research formed the base of this study. S. Vansteenberge thanks J. Van Opdenbosch and A. Ndirembako for their help collecting the stable isotope data and D. Verstraeten for the lab assistance. This research was funded by the VUB Strategic Research Funding (S. Vansteenberge), FWO Flanders (M. Sinnesael and S. Goderis), IWT Flanders (N. J. de Winter), Research grant G017217N (S. J. M. Van Malderen and F. Vanhaecke) and the Hercules Foundation (upgrade of the VUB Stable Isotope Laboratory).



**Table 1**

| Isotope | $^{25}Mg$ | $^{27}Al$ | $^{29}Si$ | $^{31}P$ | $^{34}S$ | $^{39}K$ | $^{55}Mn$ | $^{57}Fe$ |
|---|---|---|---|---|---|---|---|---|
| LOD ($\mu g\ g^{-1}$) | 4.0 | 9.0 | 100 | 1.0 | 7.0 | 7.0 | 0.08 | 4.0 |
| Isotope | $^{66}Zn$ | $^{85}Rb$ | $^{88}Sr$ | $^{89}Y$ | $^{137}Ba$ | $^{208}Pb$ | $^{232}Th$ | $^{238}U$ |
| LOD ($\mu g\ g^{-1}$) | 0.2 | 0.03 | 0.08 | 0.01 | 0.1 | 0.008 | 0.0005 | 0.0001 |

**Table 1**: Overview of limits of detection (LOD) of trace elements measured for this study using LA-ICP-MS.





**Table 2**

| Proxy | P19 | P17 | P16 |
|---|---|---|---|
| Average Layer Thickness | Thin: 0.382 mm<br>Larger variations (RSD = 28.9%) | Thick: 1.096 mm<br>Smaller variations (RSD = 6.3%) | Thick: 1.135 mm<br>Smaller variations (RSD = 9.5%) |
| $\delta^{18}O$ | Strong seasonality: tendency towards in phase correlation with $\delta^{13}C$<br>**Partially T-controlled, but other processes as well** | Weak to no seasonality: unclear relation with $\delta^{13}C$ | Weak to no seasonality: unclear relation with $\delta^{13}C$ |
| $\delta^{13}C$ | Clear $\delta^{13}C$ cycle:<br>Low $\delta^{13}C$ mostly in DCL but not always<br>**$\delta^{13}C$ driven by seasonal changes in PCP** | Clear $\delta^{13}C$ cycle:<br>Low $\delta^{13}C$ always in DCL<br>**$\delta^{13}C$ driven by seasonal changes in PCP** | Clear $\delta^{13}C$ cycle:<br>Low $\delta^{13}C$ always in DCL<br>**$\delta^{13}C$ driven by seasonal changes in PCP** |
| Mg and Sr - Ba | Good in phase correlation<br>**Mg, Sr and Ba driven by seasonal changes in PCP** | Phase relation not clear<br>**Transition period between P16 and P19 hydrological regimes** | Anti-phase correlation between Mg and Sr, Ba<br>**Seasonally occurring IDD dominates over PCP** |
| Zn, Y and Pb | Weak seasonality in Zn and Y,<br>Strong seasonality in Pb<br>**Decreased flushing, anthropogenic Pb enrichment** | Weak seasonality in Zn, Y and Pb<br>**Decreased flushing** | Very strong seasonality in Zn and Y,<br>weak seasonality in Pb<br>**Enhanced flushing** |
| U | Strong seasonality, antiphase with Mg, Sr and Ba<br>**No PCP control, scavenging** | Weak seasonality antiphase with Sr and Ba<br>**scavenging** | No seasonality |
| Remarks | Link with trace elements and layering is challenging | Link with trace elements and layering is challenging | Link with trace elements and layering is challenging |

**Table 2: Schematic overview providing the observed changes and interpretation for the different proxies of P19, P17 and P16. PCP = prior calcite precipitation, IDD = incongruent dissolution of dolomite, DCL = dark compact layers, WPL = white porous layers**








**FIGURES**

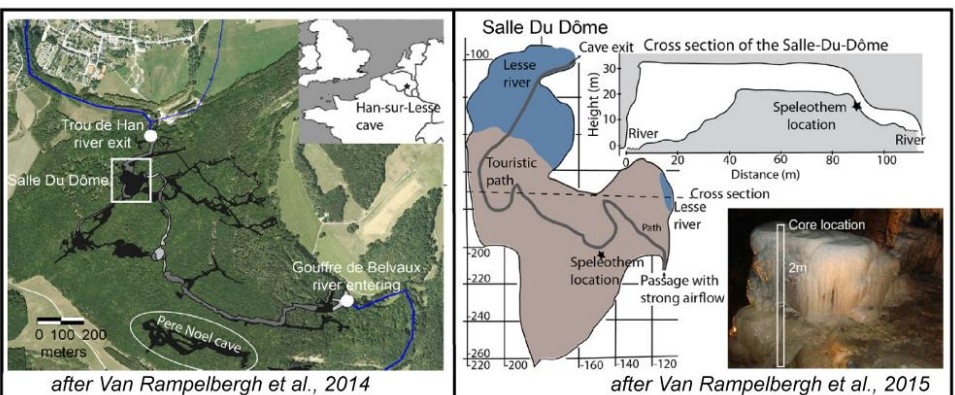

**Figure 1:** (left) Location of the Han-sur-Lesse Cave system (N50.114251, E5.203342) with the entrance
and exit of the Lesse River, the Salle-du-dome and the Père-Noël Cave. North is upwards (right) Map
showing the location of the Proserpine stalagmite within the Salle-du-Dome. The insert shows the
position of the core retrieved from the speleothem. Images adapted from Van Rampelbergh et al.
580 (2014, 2015).


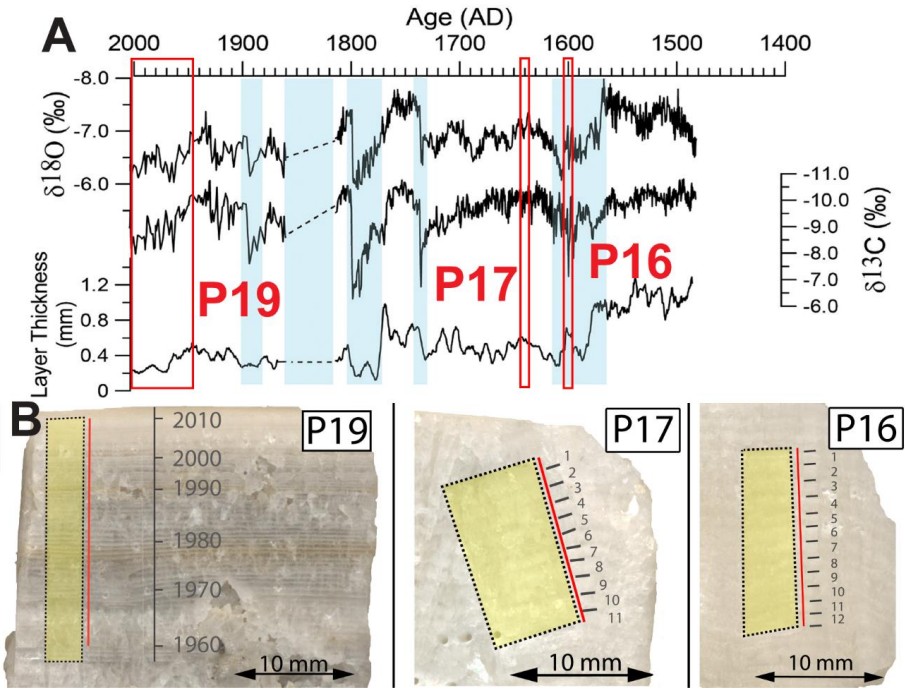

**Figure 2:** A. Overview of long (~500 yr) record of stable isotope ratios and annual layer thickness through the Proserpine speleothem measured by Van Rampelbergh et al. (2015). Red boxes indicate the locations of high-resolution transects discussed in this study. B. The three studied growth periods P19 (1960-2010 CE), P17 (1633-1644 ± 30 CE) and P16 (1593-1605 ± 30 CE). The yellow rectangles mark the sections that were drilled/sampled for $\delta^{13}C$ and $\delta^{18}O$ analysis, the red lines represent the LA-ICP-MS transects. Numbers in grey indicate the observed layer couplets.



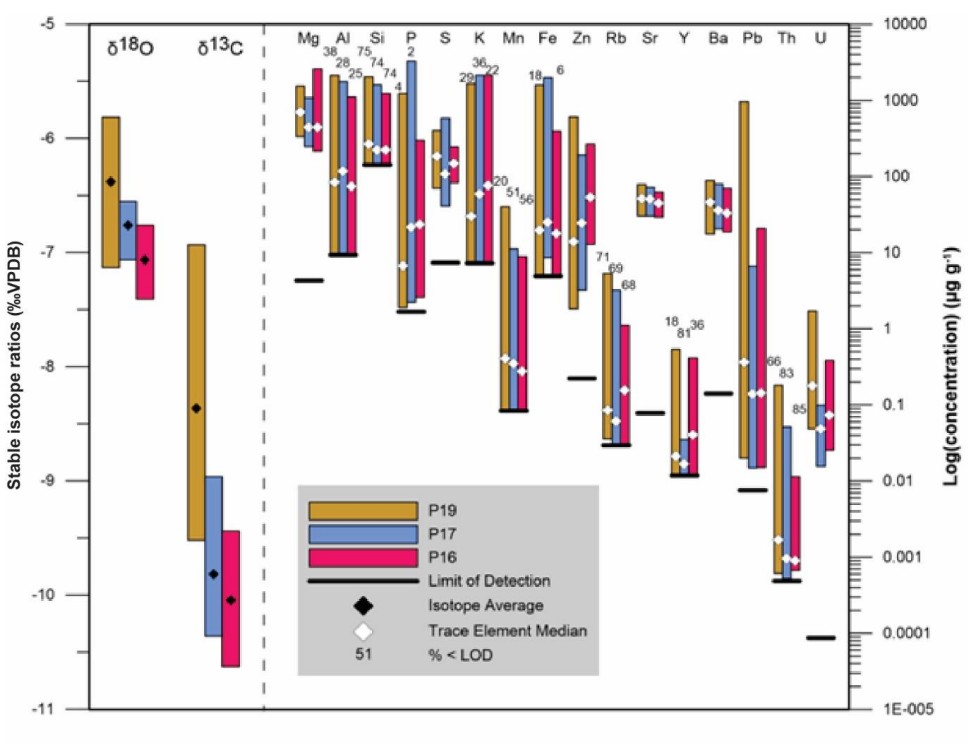

590

**Figure 3:** Ranges of the stable isotope (left) and trace element data (right). For the stable isotope
ratios, the data mark the average (black diamonds) and the standard deviation (1σ) of the
distribution. For the trace element concentrations, the boxes represent the minimum and maximum
values and the white diamonds mark the median. Numbers on top of the bars represent the
percentage of the data that is below the calculated detection limit.

596

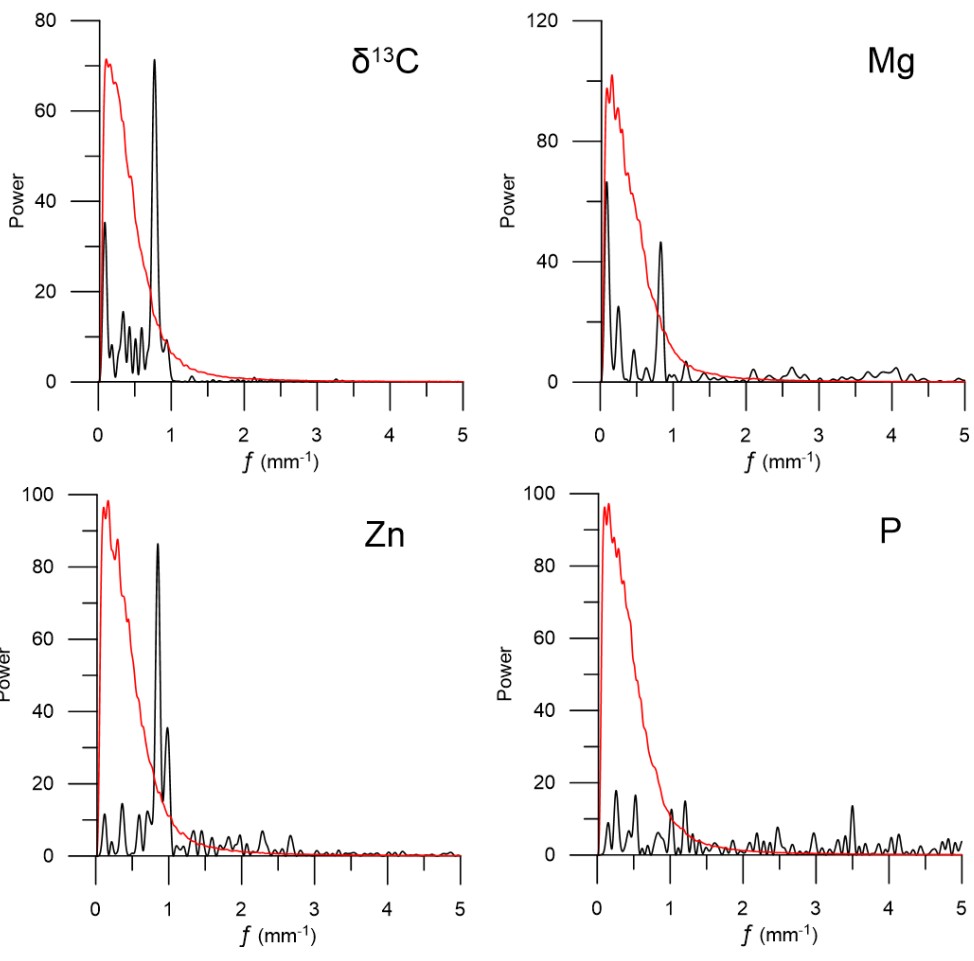

**Figure 4:** Periodograms (FFT) of δ¹³C, Mg, Zn and P measured in P16 to illustrate how the quality of a proxy to record the seasonal cycle can be studied. The red line represents the 95% confidence level. δ¹³C is taken as a reference. The periodograms include two examples of proxies with a distinct peak in the seasonal frequency band of 0.8 mm⁻¹(Mg and Zn) and one proxy with no peak in the seasonal frequency band. Periodograms for all periods are provided in the supplementary material.





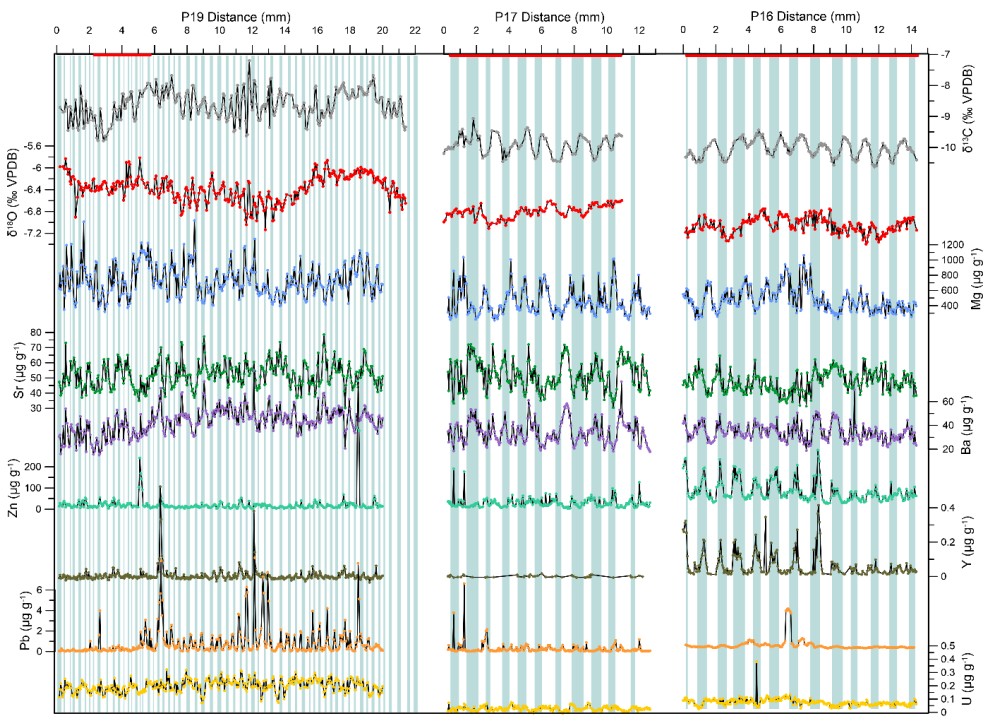

**Figure 5:** Stable isotope ratios and trace element variations plotted against distance for P19, P17 and P16. Blue bars mark the DCC laminae. The left side represents the youngest layers. All stable isotope ratios are expressed as ‰ VPDB, while trace element concentrations are reported in ppm. Red bars indicate years used for annual stack (Fig. 5 and 6).



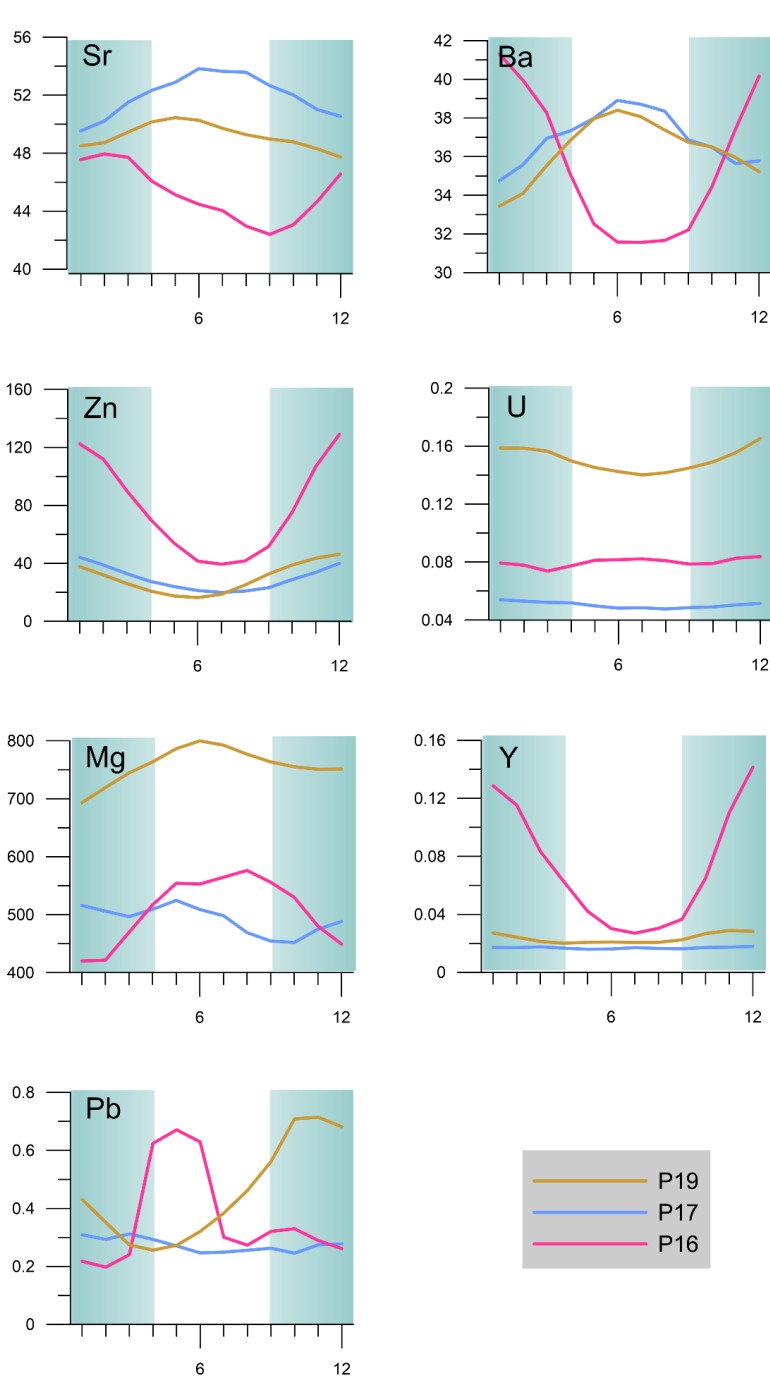


**Figure 6:** Annual stacks of the trace element proxies. Y-axis: concentrations (µg g$^{-1}$); x-axis: sub-annual
increment. For the years used, see Fig. 5.






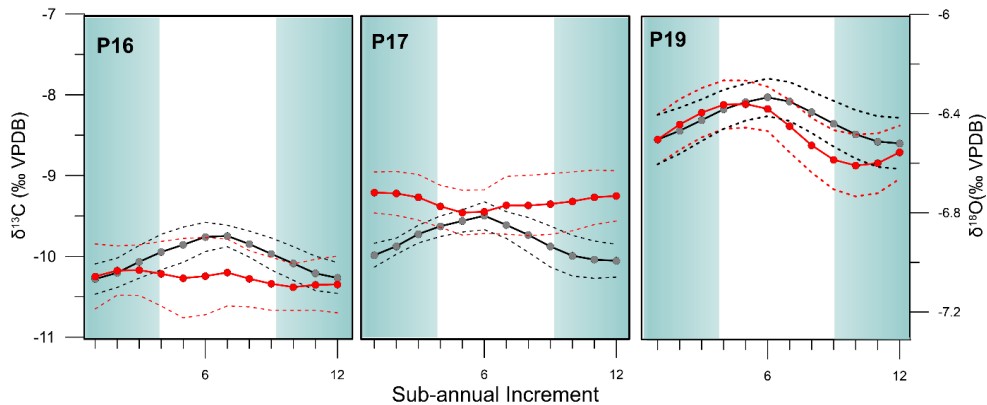


**Figure 7:** Annual stacks of δ¹³C (black) and δ¹⁸O (red). Dashed lines mark the 2σ uncertainty. The x-axis
represents one year. For the years used, see Fig. 5.

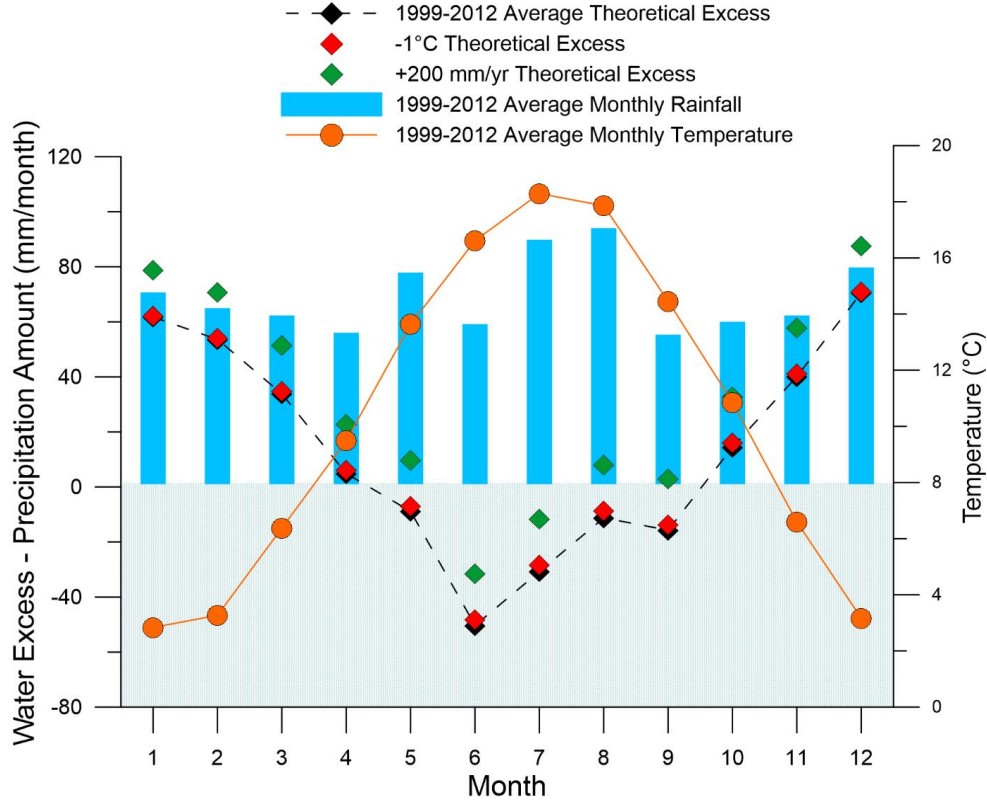


**Figure 8**: Chart showing the calculated theoretical amount of water excess calculated with the Thornthwaite equation (Thornthwaite and Mather, 1957), based on temperature and precipitation data near Han-sur-Lesse cave from 1999 to 2012 (Royal Meteorological Institute, KMI). X-axis represents the months from January to December.





623

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
