# Peer review of "Reconstructing seasonality through stable isotope and trace element analysis of the Proserpine"

_Climate of the Past, 2019_

## Referee Comment (RC1) · Robert Andrew Jamieson (Referee) · 7 Aug 2019

This is an excellent paper which pushes speleothem science forward in a significant way. The last paragraph of the conclusions in particular is an absolutely key insight which all speleothem scientists should bear in mind going forward.

In addition to vital considerations of the challenges in interpreting records within a stalagmite where the underlying cause (or causes) shift in importance through time,

the authors also present useful information about historical climate variations in central Europe.

I recommend that this manuscript should be published, and only have a few (hopefully constructive) minor comments and tweaks which I would suggest are made by the authors:

In section 2.1 (lines 92-101) the modern climate is summarised and reference made to calculated evapotranspiration effects. This information is essential, and is further discussed later on in section 5.3 where further calculations and a figure are included. However, no figure is shown here to summarise the data. This could be fixed with a reference to the subsequent figure (Figure 8, currently).

Additionally in this section, average (mean?) values are given for both rainfall and temperature. Given the main thrust of this paper is about seasonality and variability it would seem important that some discussion of the variability is also included (SDs on means, some kind of measure of mean ranges?).

Line 107: "Drip Water" within the cave is discussed. It would be useful to indicate whether this is a general cave value, or specific to the drip site where the stal in question was collected, as with flow path variations these may be different.

Line 108: Average pCO2 is given, but no clear indicator of the range. If dissolution or non-deposition are a threat to speleothem growth the maximum value is key.

Line 234: Typo, missing the word "of" at the end of the line

Line 241: (This is very nitpicky of me, and I do apologise). "the median is used instead of the average" is an odd phrasing, since the median is a form of average. I assume here that "average" refers to the mean. Here and throughout the rest of the manuscript I would ensure that either "mean" or "median" are used in the relevant places to avoid any confusion.

Line 371: Typo, "chances" should read "changes"

Section 5.3: I like this section in general, but I find the idea that a speleothem "switches" between two transfer functions to be a little simplistic. My way of thinking about it (as expressed, perhaps poorly in my 2016 paper which is cited in this manuscript) is to view correct transfer functions as a compound function with multiple terms where the weighting changes. PCP and IDD are both taking place the entire time, but the weighting shifts to make one or the other more dominant. This is perfectly compatible with the results, interpretation and conclusions drawn in this paper. The point stands that one can't simply assume a transfer function based on very short-term monitoring, because there may be a currently low weighted additional term in that function which becomes significant in other time periods. Rather than abrupt switches from one to the other (such as may sometimes be the case, e.g. with flowpath activation) the more common result may be a gradual transition. The author's shouldn't feel that this comment requires a change to the text, I just think they should consider this as an alternative way of describing the phenomena.

Line 450: Supp. Mat. Fig. 1 is referred to. Incorrect? S1 is a set of frequency analysis plots.

Figure 5: In Figure 5 the P16 section of the record appears to undergo a transition at the 9mm mark where the magnitude of multiple trace element cycles markedly alters. Y, Zn and Mg all seem to increase in cycle maximum magnitude by almost double. This sub-sub-section variability isn't discussed in the text. It probably should be at least mentioned in the results section.

---

## Referee Comment (RC2) · Anonymous Referee #2 · 29 Aug 2019

The manuscript of Vansteenberge et al presents a number of geochemical analyses across three short time intervals from the well-studied Proserpine stalagmite core from Han-sur-Lesse cave, Belgium. The analyses are very high-resolution, multi-proxy, and high quality, and yield interesting insights into the climate particular to those intervals. The manuscript is very well-written and high-quality throughout, and I recommend publication following the authors' consideration of the following points.

[Figure]

In particular, I suggest that the authors investigate ways of strengthening their discussion regarding the palaeoclimate implications of their results, which I think at the moment are too limited. An enhanced discussion of how intervals of high-resolution data derived from stalagmites can help understand climate further back in time (e.g., beyond the instrumental era) would help increase the impact of the research.

More details regarding the U-Th dating should be included in the main text, including a better description of how U-Th dating and layer counting were combined (line 161). I appreciate that this was done in previous publications and is in the SOM, but a short review (two or three sentences) outlining how layer counting and the U-Th were combined would be useful in the main text. Similarly, the U-Th-derived growth rate does not really feature, but it would be useful to compare to the presumed annual cycle wavelengths; they should be broadly similar (e.g., near lines 271-272).

I think that there must be a better way of referring to the time intervals other than the 'P16, P17, and P19'. I did not see any reason stated for why the intervals are named this, and I assume there must also be other intervals (e.g., 'P18') that also exist (maybe as slabs or pencils? Although Fig S5 doesn't imply this) but are not discussed. It almost seems that the number is linked to the century of growth, but of course this doesn't quite work ('P16' is partly in the 16th Century, 'P17' is in the 17th Century, but 'P19' is in the 20th Century). Perhaps simply changing 'P19' to 'P20' and stating that the number corresponds broadly to the century C.E. might help. I suspect that the numbers represent a label of some sort, in which case the authors may not want to change these for bookkeeping reasons. In which case, I suggest that i) somewhere on lines 75-77 it would be useful to explain why the intervals are given these labels, and ii) it might also be useful to occasionally remind the reader what these labels refer to. If changing the labels isn't possible, this will help reader keep track of the time intervals represented.

The authors should consider a plot similar to the one that they refer to from Jamieson et al., 2016 (lines 451-455). The approach used in Jamieson et al.'s Figure 6 could

really help to differentiate some of the processes particular to the three intervals discussed here. Perhaps carbon isotopes versus Mg would yield interesting insights into the different seasonal cycles inherent to the three intervals. The composite monthly geochemical proxy values shown in Figs 6 and 7 could be plotted as X and Y-axes, with the months labelled for all three intervals.

I am surprised that P does not show an annual cycle. Although the explanation suggested in lines 321-323 is certainly possible (albeit vague), it is still surprising that with a clear 'autumnal flush' of soil-derived material that P should remain unaffected. Is it possible that there are some P cycles, but that these are discontinuous so that the FFT results suggest that there are no annual cycles? Adding P to Figure 5 could help the reader can evaluate this.

Technical points: Line 15: The stalagmite does not have to be annually laminated to reconstruct monthly-scale climate. Rephrase, maybe emphasising the other benefits of annual laminations. 24: PCP will probably occur all the time, so best to say something like '...enhanced PCP occurs during...' 29: What is it in the trace element concentration profiles that reflects increased recharge? Increased specificity will help build your case early on. 36: What about other factors not discussed here? P and large organic acids in particular can depress calcite growth rates and influence partition coefficients, particularly for Sr. 42: Again, annual layering is not inherently linked to our ability to discern seasonality. Also, reconstructions can reach monthly- or even daily-scale using the right techniques, so this should be rephrased. 76: I'm sure that there is a good reason, but it would reduce confusion to explain why these intervals are called 'P16' etc. Are these the names of the stalagmite slabs or pencils? 126: I generally disagree with the requirement for stalagmites to necessarily be at isotopic equilibrium. They grow via kinetic processes so some kinetic fractionation is inevitable, and if there is more it usually enhances the primary climate signal. So I am not concerned if there are occasional intervals with more or less disequilibrium fractionation within the context of the current study. 144: 'number' instead of 'amount' 156: I seem to recall that a core

from the same stalagmite contained petrographic evidence for a reduction in visitation to the cave during both world wars (Verheyden et al., 2006). I assume that this is the same core - is the chronology presented here still consistent with this interpretation? If so, it might be useful to state here because it would help confirm the chronology. 163: 'coupled to a Leica GZ6 microscope' 190: 'laser points'? 191: 'because' 193: elements are lowercase – here and throughout 239: 'mean' instead of 'average' here and in most other occurrences (other than when not referring to a statistical mean, e.g., 'the average cave is. . .' etc.) 271: How does this compare to the U-Th derived growth rate? 295: '. . .the Han-sur-Lesse Cave. . .' 457: Is deposition on the roof (i.e., on the stalactite) not enough? Is another void space really necessary? If there is a large stalactite over the Prosperpine stalagmite, deposition on this might be enough to cause the observed PCP signal (I agree that it is PCP). 477: I agree with how the authors handle evapotranspiration, but I'd like them to consider the possibility that the concept isn't really relevant in highly karstified regions. If a summer month experiences an intense rain event, the rain may be channelled into a doline and into the subsurface before 'evapotranspiration' has a chance to really occur. If the Proserpine stalagmite is fed by a rapid drip with high variability, it is possible that the whole evapotranspiration concept does not apply. Hess and White (in the book "Karst Hydrology: Concepts from the Mammoth Cave area") suggest a 13% decrease in the relevance of evapotranspiration in karst regions, but I suspect it could be even more in certain situations. The authors approach is correct and is what is conventionally done, so if reducing the amount of summer evapotranspiration is useful they may want to consider this possibility – otherwise I'll leave it up to the authors whether or not they want to rephrase Conclusions: As mentioned above, I think that it would be beneficial for the authors to conclude with a greater focus on the implications for climate science. It is impractical to perform an LA-ICPMS analysis across an entire meter long stalagmite. Does this research provide any guidance as to how short high-res intervals would be useful across very long records? I think that it does, and the authors should discuss this more than they currently do. 517: Couldn't cooler regional temp also explain the decrease in

ïĄd̂'18O? Table 2 caption labelled as Table 1

---

## Referee Comment (RC3) · Anonymous Referee #3 · 30 Aug 2019

Cp-2019-78 Review

My general comments are embedded in the specific sections pertaining to Lines in the text. Overall, I believe that the principal hypothesis has to be better supported by data and the discussion has to encompass other alternate hypotheses. . .based on the data.

Line 42: Perhaps it is best to replace "their ability to hold. . ." with: Some speleothems are characterized by distinct physical and/or geochemical layering that enables past climate and environmental reconstruction at the seasonal scale (Ref. is needed here, such as, for example: Mattey et al., 2008; Boch & Spötl, 2008 ).

Line 48: it is not entirely correct that visible laminae develop because of change in drip rate. They can also be caused by variation in cave ventilation, and, therefore, in the pH of the film of fluid driven by seasonal intensity of the degassing process, which then produces laminae consisting of compact calcite and dark, impurity rich, more porous calcite (see for example Frisia et al., 2003) or white, porous calcite (Boch et al., 2010). In particular, Boch et al (2010) suggest that drip rate influences the thickness of the laminae, whilst the fabrics reflect seasonality in degassing. True that the two may be related (degassing and fabric) but this needs to be somehow written, just to show that there is complexity in this two-fabric system. Actually, such clarification would explain best the following paragraphs.

Line 51: PCP can also occur in the epikarst, in fractures or pores, not just in the cave. So, it is not entirely appropriate to state that the ability of the cave waters to degas is increased. Infiltration waters is best.

Line 54: replace is known with "has been known"

Line 55: replace stalagmite collected with "Proserpine stalagmite, which was collected in the cave of Han-sur-Lesse and first studied by Verheiden. . .. Line 57: replace "according to" with "as inferred from U/Th dating and lamina counting between the radiometric ages (. . ..).

Line 65: there are many papers before the Regattieri et al. (2016) which demonstrated the importance of trace elements in tracing the seasonality of hydroclimate in stalagmites. Please refer to Treble et al., 2003; Fairchild and Treble (2009), Griffiths et al., (2010).

Line 70: actually, the first to use trace elements to fingerprint volcanic events were Frisia et al. (2005). The most accurate expression here would be: and can be

used to identify changes in atmospheric load of anthropogenically and volcanic derived aerosols, as well as volcanic ash

Line 71: please add after geochemical layering : "of Proserpina" line 73: replace "the relation" with its relation. Replace "this work" with the present study. Line 75: I would rephrase for clarity: To achieve….three sections of the Proserpina stalagmite, each encompassing 12 years of growth, were analysed, namely : P16 (from….to), P17 (from to ) and P 19 (from . . .to). Each section was analysed at sub-annual resolution …… this sentence is not entirely clear. Do you mean: the results from the selected sections were then applied to interpret in terms of seasonality of hydroclimate variability climate variability the Proserpina stable isotope ratio curves for the last 400 years.

Please, check if I understood correctly what was meant with your sentence. I looked at the figures . . . but I may be wrong.

Line 78 replace elemental with element

Apologies, I have become a bit annoyed at having to correct grammar and typos. . .I have started highlighting issues in yellow on the pdf to save my time.

Line 87 to 91 : local is too vague. What does it mean? Partially dolomitized limestone? As the geology is relevant to the data and discussion, a geologic section of the beds above the cave and inferred distribution of dolomite should be provided. Fig. 1 does not give any geologic information. Somehow, the text is referring to formations that are not shown, the relevance of which is, thus, unknown to the reader. In the opinion of this reviewer, this is not scientifically acceptable, as it fails to provide crucial information. Lines 95-97: Just a suggestion – it would be great help to the reader if a figure with distribution of rainfall and temperatures throughout the year (or a 30 year mean) were presented. True, we are referred to Genty &Deflandre (1998) and Genty & Quinif (1996), but still it is beneficial to have it visually in the present manuscript, even as Supporting Material. It would help understanding the significance of proxy data. Line 103: I suggest to write that it was removed in 2001. Line 108: colder than what? Line

Interactive
comment

109: what is the time scale of "temporary"? Line 114 replace "record" with values. Line 117: what does tabular mean? Better to rephrase: flat topped surface. Even if tabular was used by Rampelbergh et al. (climate of the past 2015), still the speleothem looks more a composite edifice showing, clearly, from the insert in Fig 1. From Fig. 1 (and Fig. 1b in Rampelbergh et al., 2015) Proserpina it is not a typical stalagmite, but a speleothem that formed through the coalescence of several stalagmites, because of the fast drip rate, into a stalagmitic flowstone characterized by mm- to cm- scale rimmed pools (which is also clear from Verheyden et al., 2006, Figures 1 and 2 showing cahaotic "pool like" deposits). Which hints to the fact that the feeding drip points may be multiple, shift in time and drip rates may have been extremely variable because of rupture of feeding straws (Verheiden et al, Fig 1). And, in fact, the sentence continues by stating that it is fed by a drip "flow". Thus, I would call it a stalagmitic flowstone. Line 122. State entirely what DCL and WPL are the first time. In any case. Line 156: The way the sentence is written, it would seem that there 50 years of uncertainty > I suppose that "at" should be replaced by from 1960 (to 2000). Line 231 It is unclear to this reviewer why line P17 was taken at an angle with crystal growth. In Fig. 2 B one can see that the directions of columnar crystal growth at the core of the photo is from bottom to top. The P17 was taken laterally. It is unclear, from the photo, if it is meant to be perpendicular to an outer layer , roughly at 45 degrees from the "page vertical direction). Yet, from the "blurred" image I could retrieve, it seems that the fabric is more complex, there could be more "mosaic-type" crystals as in P 19. Please explain the direction of P17, which may actually explain the isotope ratios data in Fig 2, specifically the abrupt shifts. I suggest the Authors to provide a better fabric control for their data. His may also be the reason why the thickness are different for P16 and 17 relative to 19. In P 19 it is clear that the transect cross cuts laminae perpendicularly. Line 234. Perhaps there is an "of" missing? Line 288. Perhaps better to expand for those who are not familiar. Rather than "same samples" elaborate: whilst stable isotope ratios are measured on powders drilled... trace elements are...thus the exact location of the analyses in the same sample may not coincide.... Line 297. What is intended here for mineral? Dissolved ions? And what is intended for organic matter? Please explain. Have any analyses for the nature of the DOC in the H sur L waters been carried out? The type of Organics? LMW-A, LMW-N, building blocks etc? Works by Hartland et al. (2012), Rutlidge et al. (2014) are showing a connection between the NOM and trace element transport in cave waters (and trace element content in stalagmites). Thus, in order to better understand the trace element variability, a knowledge of DOC in H sur L waters is needed (see the speculation at Line 8. This can be solved by having a characterization of NOM in the feeding waters). Line 316 to 323. The discussion about P becomes highly speculative and citing Frisia et al (2012) is, here, not the case, as the case-study in the cited paper is from locations where P is known to be present in the host rock and/or is related to stromatolite-like features. Unless the Authors show petrographic features that resemble microbially induced ones, the hypothesis has no grounds. And the same applies for the host rock, Authors do not provide a geochemical characterization of the rock. Hence, Authors simply do not know where P may come from and why it behaves the way it does. This is why they need to provide a DOC characterization of the dripwater. As there are organic acids that favour the mobilization of P. Each cave has its own environment and diverse types of soil. Also: Vegetation dieback for P was hypothesized in a seminal paper by Treble et al (2003), which I believe should be cited, as it provides a diverse context that that of Baldini et al (2002).

Lines 339 to 393 The issue here is that the different behaviour from P19, P16 and P17 is that P19, from Fig 2, is the only clear transect that is perpendicular to parallel growth layers that are perfectly flat. The delta 13C values and flat tops with a seemingly predominant compact fabric indicate that P19 formed from a thin film of fluid that resided for a longer time (than P17 and 16) at the top surface of the growing stalagmite. This allowed prolonged degassing. At this stage, the correlation between delta 13 C, Mg and Sr may be slightly different than just PCP, the....delta 13 C may be also influenced by in- cave degassing, but all processes lead to the same conclusion: less infiltration. For the antiphase correlation between Mg and Sr, the Authors speculate about a role for dolomite dissolution, albeit the dolomite bodies are not presented in a geologic section. If this were the case, it should hold also forP19, unless the hydrological pathway changed completely. But there is another explanation, for which the Authors should consult Rutlidge et al (2014). The Authors should actually analyse other elements, such as Al, K, Si and check their behaviour in relation to Mg and Sr. Works that use Sr isotopes have highlighted that Sr may have a twofold provenance: from soil and from rock (Belli et al.,2017). Mg has also a similar twofold provenance (Rutlidge et al., 2014). Now, if Sr had provenance from dolomite and limestone, the Authors should explain why the overall Sr values in P19, 16 and 17 are similar. If Sr in P19 came only from limestone, why is it that the baseline is the same (about 60 ppm?). Also, the Mg concentration in P 19 is overall higher than in P16 and 17, which should not be expected if the hydrological pathway was in a pure limestone, or dolomitized limestone. So, the actual possibility is that something changed between 16 and 17, rather than "somewhat" between P16 and P19., but has nothing to do with dolomite (or little to do with it). It is the Y and Zn cycles that actually point to that. It could be soil. The other problem is that the Author cite shales. And shales may be rich in Sr. And Zn and Y Thus, trace elements and/or Sr isotopes that point to provenance of Sr (soil or shales) would give robustness to the speculation. Line 415Incongruent dolomite dissolution is not really demonstrated by the data, as I mentioned above, there should be a change in the mean Mg and Sr values from P19 and P17/16, there is not. The baseline is similar. Also, in P17 and 16 the Mg is not always co-varying with the delta13C (Fig.5). If IDD were acting, I would expect that higher Mg coincided with more positive delta 13C, because of dolomite dissolution. But when the Mg is high and delta 13C is more negative, that there is an issue. Unless Mg comes from the soil, or the dolomitized shales have a low delta13C because they are rich in organic matter (I suppose they are dark, but I may be mistaken). To support the IDD hypothesis, I sugget that the Authors provide a geochemical composition for the limestones, the shales and the dolomite, which would help understanding the effects of the processes they invoke (PCO vs IDD) Line 448 I do not understand why a change in "cave morphology" is cited here. IWhat would be important, from the data presented by the Author, is a change in ENVIRONMENTAL

parameters. Not necessarily climate. This reviewer admits a poor knowledge of the studied region, but being in Belgium, I suppose that humans have somewhat impacted the land. Even if it does not appear "visible", land use (tree cutting, particularly in the LIA, when people had to find wood somewhere….) may have altered the hydrological pathways, changed soil (uprooting). These options…of anthropogenic disturbance, need to be investigated also for P 17 and 16, not just P19. Line 427: may also indicate that soil disturbance/forest disturbance above the cave resulted in faster infiltration for P19, because of less soil humidity retention. And, obviously, the rise in temperature in the last 150 years related to the end of LIA, industrial "revolution" and increase in GHG in the atmosphere. Line 473-489 This part of the discussion does not account for land use. Because the data are discussed with the assumption of PCP versus IDD. But this reviewer fails to see unequivocal proof of IDD from the trace element data. Lines 490-517 Actually, the best piece of information in favour of higher recharge in the 17th century is the lamina thickness, given that the delta13C values seem similar over the three periods. The delta 18O could reflect cooler air masses in the 17th century, or a diverse provenance. So it may not be a n unequivocal marker of enhanced rainfall in winter. But the thicker laminae in a cold period need to have water supply and an efficient soil/microbes/fungal hyphae. The marked seasonality could well be a marker of snowmelt, which should be considered as hypothesis. Actually, if the bulk of infiltration in P16 and 17 came at snowmelt rather than in Autumn (as I suppose there is no historical record of rainfall patterns for the 17th Century), then the behaviour of P may be explained…there is no vegetation dieback in Spring.

Hope these comments help.

Please also note the supplement to this comment:
https://www.clim-past-discuss.net/cp-2019-78/cp-2019-78-RC3-supplement.pdf
* * *
[Figure]

**Supplement:**

[revised manuscript text omitted]

---

## Author Response (AR1)

Dear Climate of the Past Editorial Board, Dear reviewers,

On behalf of all the authors, I would like to express my gratitude to the three reviewers who have published comments on our work in the interactive discussion. Below, we provide a point-by-point reply to all questions and remarks raised by our reviewers and provide suggestions to how we can improve our manuscript taking into account the comments of our reviewers. We reiterate comments by reviewers in italics and provide our answers (in red) directly below the comments.

**Reviewer 1 (Robert Andrew Jamieson)**

*This is an excellent paper which pushes speleothem science forward in a significant way. The last paragraph of the conclusions in particular is an absolutely key insight which all speleothem scientists should bear in mind going forward.*

*In addition to vital considerations of the challenges in interpreting records within a stalagmite where the underlying cause (or causes) shift in importance through time, the authors also present useful information about historical climate variations in central Europe*

*I recommend that this manuscript should be published, and only have a few (hopefully constructive) minor comments and tweaks which I would suggest are made by the authors:*

*In section 2.1 (lines 92-101) the modern climate is summarised and reference made to calculated evapotranspiration effects. This information is essential, and is further discussed later on in section 5.3 where further calculations and a figure are included. However, no figure is shown here to summarise the data. This could be fixed with a reference to the subsequent figure (Figure 8, currently).*

We will refer to current figure 8 (moving it forward in the manuscript) displaying a climatogram of the modern climate

*Additionally in this section, average (mean?) values are given for both rainfall and temperature. Given the main thrust of this paper is about seasonality and variability it would seem important that some discussion of the variability is also included (SDs on means, some kind of measure of mean ranges?).*

We agree that this is a good idea and will add a sentence describing the present-day climate in terms of seasonal variability.

*Line 107: "Drip Water" within the cave is discussed. It would be useful to indicate whether this is a general cave value, or specific to the drip site where the stal in question was collected, as with flow path variations these may be different.*

Drip waters were sampled by Van Rampelbergh (2014) directly at the site of the Proserpine speleothem and are therefore representative for this study. We will make mention of this in the revised manuscript version.

*Line 108: Average pCO2 is given, but no clear indicator of the range. If dissolution or non-deposition are a threat to speleothem growth the maximum value is key.*

The $pCO_2$ value at the site of the Proserpine varies through the year (see Van Rampelbergh et al., 2014). We will make mention of the range of values for clarity.

*Line 234: Typo, missing the word "of" at the end of the line*

We will add the word "of" here.

*Line 241: (This is very nitpicky of me, and I do apologise). "the median is used instead of the average" is an odd phrasing, since the median is a form of average. I assume here that "average" refers to the mean. Here and throughout the rest of the manuscript I would ensure that either "mean" or "median" are used in the relevant places to avoid any confusion.*

We agree that this is confusing and will replace any mention of "average" in the manuscript with the word "mean" to avoid this confusion. This was also noted by other reviewers.

*Line 371: Typo, "chances" should read "changes"*

We will correct this in the revised version.

*Section 5.3: I like this section in general, but I find the idea that a speleothem "switches" between two transfer functions to be a little simplistic. My way of thinking about it (as expressed, perhaps poorly in my 2016 paper which is cited in this manuscript) is to view correct transfer functions as a compound function with multiple terms where the weighting changes. PCP and IDD are both taking place the entire time, but the weighting shifts to make one or the other more dominant. This is perfectly compatible with the results, interpretation and conclusions drawn in this paper. The point stands that one can't simply assume a transfer function based on very short-term monitoring, because there may be a currently low weighted additional term in that function which becomes significant in other time periods. Rather than abrupt switches from one to the other (such as may sometimes be the case, e.g. with flowpath activation) the more common result may be a gradual transition. The author's shouldn't feel that this comment requires a change to the text, I just think they should consider this as an alternative way of describing the phenomena.*

We think this is a valid comment and thank the reviewer for better explaining his point made in the paper we cite in this section. We will very likely modify this section in reply to comments from the other reviewers and to leave room in the discussion for alternative hypotheses (such as the land use change proposed by Reviewer 3, which is indeed a plausible alternative).

*Line 450: Supp. Mat. Fig. 1 is referred to. Incorrect? S1 is a set of frequency analysis plots.*

Correct, this reference should be to Figure 2 (which we moved from the supplement in an earlier version). We will update this reference.

*Figure 5: In Figure 5 the P16 section of the record appears to undergo a transition at the 9mm mark where the magnitude of multiple trace element cycles markedly alters. Y, Zn and Mg all seem to increase in cycle maximum magnitude by almost double. This sub-sub-section variability isn't discussed in the text. It probably should be at least mentioned in the results section.*

Agreed, we have overlooked this apparent change in the current discussion of the data and will make mention of it at least in the results of the new version.

**Reviewer 2 (Anonymous)**

*The manuscript of Vansteenberge et al presents a number of geochemical analyses across three short time intervals from the well-studied Proserpine stalagmite core from Han-sur-Lesse cave, Belgium. The analyses are very high-resolution, multi-proxy, and high quality, and yield interesting insights into the climate particular to those intervals. The manuscript is very well-written and high-quality throughout, and I recommend publication following the authors' consideration of the following points.*

*In particular, I suggest that the authors investigate ways of strengthening their discussion regarding the palaeoclimate implications of their results, which I think at the moment are too limited. An enhanced*

*discussion of how intervals of high-resolution data derived from stalagmites can help understand climate further back in time (e.g., beyond the instrumental era) would help increase the impact of the research.*

This is a helpful suggestion that will indeed make our manuscript stronger. We will implement it by adding a short paragraph at the end of our discussion in which we more clearly explain the implications of high-resolution data for climate reconstructions from speleothems on different timescales.

More specifically, we will emphasize that the expression of trace element records should not be seen as a result of a constant transfer function (see Reviewer 1's comment on our section 5.3). Instead, we propose that the application of high-resolution, multi-proxy transects placed on strategic places along the length/growth axis of a speleothem may be used to check for changes in the response of these proxies to long-term trends and with respect to each other. These changes have implications for the interpretation of the longer, lower resolution proxy records, highlighting the importance of including multiple proxies (e.g. trace element in combination with stable isotope, sedimentary and/or crystallographic analyses) as well as multiple sampling densities (high- and low resolution sampling) to reliably interpret speleothem archives in terms of climate and environmental evolution.

*More details regarding the U-Th dating should be included in the main text, including a better description of how U-Th dating and layer counting were combined (line 161). I appreciate that this was done in previous publications and is in the SOM, but a short review (two or three sentences) outlining how layer counting and the U-Th were combined would be useful in the main text. Similarly, the U-Th-derived growth rate does not really feature, but it would be useful to compare to the presumed annual cycle wavelengths; they should be broadly similar (e.g., near lines 271-272).*

We will add a few sentences about our age model to our methods section and discuss the (differences in) growth rates derived from the model in our discussion.

The number of counted layer couplets over the annually layered 500-years determined the seasonal character of the layers and demonstrated that two layers (one dark, one light) were deposited per year. The number of years obtained by layer counting between two U-Th datings was compared with the number of years suggested by the U-Th ages. We combine results of both independent dating methods to produce the final age model.

*I think that there must be a better way of referring to the time intervals other than the 'P16, P17, and P19'. I did not see any reason stated for why the intervals are named this, and I assume there must also be other intervals (e.g., 'P18') that also exist (maybe as slabs or pencils? Although Fig S5 doesn't imply this) but are not discussed. It almost seems that the number is linked to the century of growth, but of course this doesn't quite work ('P16' is partly in the 16th Century, 'P17' is in the 17th Century, but 'P19' is in the 20th Century). Perhaps simply changing 'P19' to 'P20' and stating that the number corresponds broadly to the century C.E. might help. I suspect that the numbers represent a label of some sort, in which case the authors may not want to change these for bookkeeping reasons. In which case, I suggest that i) somewhere on lines 75-77 it would be useful to explain why the intervals are given these labels, and ii) it might also be useful to occasionally remind the reader what these labels refer to. If changing the labels isn't possible, this will help reader keep track of the time intervals represented.*

This is a useful suggestion and we agree that the naming of the slabs might be confusing to the reader. We like the suggestion of changing 'P19' to 'P20' to refer (broadly) to the centuries and will follow this suggestion in our revised manuscript.

*The authors should consider a plot similar to the one that they refer to from Jamieson et al., 2016 (lines 451-455). The approach used in Jamieson et al.'s Figure 6 could really help to differentiate some of the processes particular to the three intervals discussed here. Perhaps carbon isotopes versus Mg would yield*

*interesting insights into the different seasonal cycles inherent to the three intervals. The composite monthly geochemical proxy values shown in Figs 6 and 7 could be plotted as X and Y-axes, with the months labelled for all three intervals.*

We like this suggestion for a cross plot of mean monthly changes in different proxies and will try to implement it in the revised version. We will try to focus on Mg, Sr and d13C for this plot as those are the proxies we discuss most in combination in our discussion.

*I am surprised that P does not show an annual cycle. Although the explanation suggested in lines 321-323 is certainly possible (albeit vague), it is still surprising that with a clear 'autumnal flush' of soil-derived material that P should remain unaffected. Is it possible that there are some P cycles, but that these are discontinuous so that the FFT results suggest that there are no annual cycles? Adding P to Figure 5 could help the reader can evaluate this.*

We will add the P records to Figure 5 to demonstrate that, while there is definitely some variability in P over the records, this variability is not very periodic in character and does not follow the same pattern as the other soil-derived elements (raw data is also given in supplementary tables).

Surprisingly, no seasonal changes were observed in P-content. The difference between records of P and other soil-derived elements (e.g. Zn and Y; which exhibit clear seasonality) is difficult to explain. The finding is, however, in agreement with minimal seasonal variability observed in $\delta^{18}O$ and $\delta D$ values of cave water (Van Rampelbergh et al., 2014). This suggests that seasonal changes in the epikarst, linked to water availability, were dominant over seasonal processes related to surface (soil) processes. Another explanation might be that the limit of detection of P in our LA-ICP-MS data is higher relative to the measured values than that of other elements of interest (e.g. Mg, Zn and Sr), causing higher analytical noise on the P record compared to the other trace element records. The reason might be that P measurements are sensitive to interferences in a Ca-rich matrix (such as calcium carbonate in speleothems).

*Technical points:*

*Line 15: The stalagmite does not have to be annually laminated to reconstruct monthly-scale climate. Rephrase, maybe emphasising the other benefits of annual laminations.*

We will rephrase this to make it more general and then add one additional sentence explaining the specific advantage of an annually layered speleothem.

*24: PCP will probably occur all the time, so best to say something like ': : :enhanced PCP occurs during: : :'*

We will change this and mention "enhanced" PCP.

*29: What is it in the trace element concentration profiles that reflects increased recharge? Increased specificity will help build your case early on.*

We will be more specific here and mention already in the abstract that higher peaks in soil-derived elements (e.g. Zn and Y) and lower host-rock derived elements (e.g. Mg, Sr, Ba) point towards lower residence times in the epikarst and higher flushing rates during the 17[th] century compared to present-day conditions.

*36: What about other factors not discussed here? P and large organic acids in particular can depress calcite growth rates and influence partition coefficients, particularly for Sr.*

As we will show when we add P records to Figure 5 (as discussed above), P does not show the annual variability seen in other trace elements and the P records are not significantly different between the different intervals. We will make brief mention of this record in the results but will only briefly discuss processes based on the P records, because it does not show a clear correlation with what's going on in the other proxy records (see above).

*42: Again, annual layering is not inherently linked to our ability to discern seasonality. Also, reconstructions can reach monthly- or even daily-scale using the right techniques, so this should be rephrased.*

We will rephrase this.

*76: I'm sure that there is a good reason, but it would reduce confusion to explain why these intervals are called 'P16' etc. Are these the names of the stalagmite slabs or pencils?*

As mentioned above, we will implement the suggestion to rename 'P19' to 'P20' and motivate the names by stating that they derive from the (approximate) centuries in which the intervals grew.

*126: I generally disagree with the requirement for stalagmites to necessarily be at isotopic equilibrium. They grow via kinetic processes so some kinetic fractionation is inevitable, and if there is more it usually enhances the primary climate signal. So I am not concerned if there are occasional intervals with more or less disequilibrium fractionation within the context of the current study.*

We would like to emphasize that we never in the text stated that speleothems are necessarily in isotopic equilibrium. Since our sentence around line 125-126 seems to make this impression on the reader we will rephrase it to clarify that this is not what we mean.

*144: 'number' instead of 'amount'*

Agreed, we will correct this.

*156: I seem to recall that a core C3 from the same stalagmite contained petrographic evidence for a reduction in visitation to the cave during both world wars (Verheyden et al., 2006). I assume that this is the same core - is the chronology presented here still consistent with this interpretation? If so, it might be useful to state here because it would help confirm the chronology.*

This is indeed the same core through the same speleothem, and the chronology presented in this study remains consistent with this previous interpretation. We will reiterate it here (referring to Verheyden et al., 2006) to help give our chronology more confidence.

*163: 'coupled to a Leica GZ6 microscope'*

We will correct this typo.

*190: 'laser points'?*

We will rephrase to "laser pits" or "laser spots" to clarify.

*191: 'because'*

This will be corrected

*193: elements are lowercase – here and throughout*

We will correct this throughout the manuscript.

*239: 'mean' instead of 'average' here and in most other occurrences (other than when not referring to a statistical mean, e.g., 'the average cave is: : :' etc.)*

Similar to a previous comment by Reviewer 1, we will use "mean" throughout the text.

*271: How does this compare to the U-Th derived growth rate?*

See earlier comment: We will describe and discuss the results of our age model in more detail in the revised version and compare estimated growth rates with the thicknesses of the layers to lend more confidence to our interpretation of annual cyclicity.

295: ': : :the Han-sur-Lesse Cave: : :'

We would like to keep this sentence as it is, referring to this monitoring study specifically as done in another part of the cave system compared to the previously cited studies by Van Rampelbergh et al., which were done at the Proserpine growth site to differentiate between these studies.

457: Is deposition on the roof (i.e., on the stalactite) not enough? Is another void space really necessary? If there is a large stalactite over the Prosperpine stalagmite, deposition on this might be enough to cause the observed PCP signal (I agree that it is PCP).

This might indeed be the case and we will specifically mention this possibility here and state that voids in the epikarst are not strictly necessary as an explanation.

477: I agree with how the authors handle evapotranspiration, but I'd like them to consider the possibility that the concept isn't really relevant in highly karstified regions. If a summer month experiences an intense rain event, the rain may be channelled into a doline and into the subsurface before 'evapotranspiration' has a chance to really occur. If the Proserpine stalagmite is fed by a rapid drip with high variability, it is possible that the whole evapotranspiration concept does not apply. Hess and White (in the book "Karst Hydrology: Concepts from the Mammoth Cave area") suggest a 13% decrease in the relevance of evapotranspiration in karst regions, but I suspect it could be even more in certain situations. The authors approach is correct and is what is conventionally done, so if reducing the amount of summer evapotranspiration is useful they may want to consider this possibility – otherwise I'll leave it up to the authors whether or not they want to rephrase

This is a valid point and we will add a short (1-2 sentence) discussion of the relevance of evapotranspiration in karst systems (including reference to the suggested citation) for sake of completeness. However, we still believe that evapotranspiration is an important process in this case. As shown in Figure 8 (which will be moved up in the manuscript, see other comments), seasonality in precipitation is rather limited in the area, except for a slight increase in the summer months, as discussed. We must therefore conclude that the large seasonal changes we observe in our data, which are controlled by seasonal changes in drip rate (as demonstrated by Genty and Deflandre, 1998), are driven for a large part by seasonal changes in evapotranspiration. We will clarify this line of reasoning in the revised discussion.

Conclusions: As mentioned above, I think that it would be beneficial for the authors to conclude with a greater focus on the implications for climate science. It is impractical to perform an LA-ICPMS analysis across an entire meter long stalagmite. Does this research provide any guidance as to how short high-res intervals would be useful across very long records? I think that it does, and the authors should discuss this more than they currently do.

This is a very helpful suggestion and we will make sure to add a paragraph at the end of the discussion (pre-conclusion) or in the conclusion in which we elaborate more on the implications of our work for similar studies. In this paragraph we will give some recommendations on how to optimize LA-ICP-MS measurements to obtain chemical information on different time scales through a speleothem, combining high resolution intervals with longer, lower resolution transects.

517: Couldn't cooler regional temp also explain the decrease in δ18O?

They could in theory explain some variation in d18O, but only if these cooler temperatures resulted in cooler cave temperatures as well, and only if fractionation was in equilibrium with ambient air, which is not certain for the 17th century part of the Proserpine. In addition, the local temperature trend of ~1°C trend is not visible in the long-term d18O record of Figure 2, suggesting that other factors have a stronger influence on the oxygen isotope fractionation in this speleothem (e.g. the d18O of precipitation). Nevertheless, we will add a sentence highlighting this possibility since we cannot exclude it.

*Table 2 caption labelled as Table 1*

We could not find this typo in the version we received directly after submission from the system. In our version, Table 2 caption is correctly labeled also in the CP discussions PDF. We will make sure the labelling remains correct in the revised version.

**Reviewer 3 (Anonymous)**

*Cp-2019-78 Review*

*My general comments are embedded in the specific sections pertaining to Lines in the text. Overall, I believe that the principal hypothesis has to be better supported by data and the discussion has to encompass other alternate hypotheses: : :based on the data.*

*Line 42: Perhaps it is best to replace "their ability to hold: : :" with: Some speleothems are characterized by distinct physical and/or geochemical layering that enables past climate and environmental reconstruction at the seasonal scale (Ref. is needed here, such as, for example: Mattey et al., 2008; Boch & Spötl, 2008 ).*

We will rephrase the sentence following the reviewer's suggestion and include the suggested citations.

*Line 48: it is not entirely correct that visible laminae develop because of change in drip rate. They can also be caused by variation in cave ventilation, and, therefore, in the pH of the film of fluid driven by seasonal intensity of the degassing process, which then produces laminae consisting of compact calcite and dark, impurity rich, more porous calcite (see for example Frisia et al., 2003) or white, porous calcite (Boch et al., 2010). In particular, Boch et al (2010) suggest that drip rate influences the thickness of the laminae, whilst the fabrics reflect seasonality in degassing. True that the two may be related (degassing and fabric) but this needs to be somehow written, just to show that there is complexity in this two-fabric system. Actually, such clarification would explain best the following paragraphs.*

This is a useful suggestion and we will implement this nuance in our description of the causes of annual lamination in speleothems in general, including the suggested references. Note that in our original sentence we tried to cover changes in cave ventilation in our mention of "cave climatology" but we understand that this is a bit vague and that it would be better to give a more detailed explanation since the annual laminations are of central importance to our manuscript. We thank Reviewer 3 for his suggestions that will help us do so.

*Line 51: PCP can also occur in the epikarst, in fractures or pores, not just in the cave. So, it is not entirely appropriate to state that the ability of the cave waters to degas is increased. Infiltration waters is best.*

We will rephrase and change "cave waters" to "infiltration waters" to clarify.

*Line 54: replace is known with "has been known"*

We will rephrase this.

*Line 55: replace stalagmite collected with "Proserpine stalagmite, which was collected in the cave of Han-sur-Lesse and first studied by Verheiden: : :.*

We will make the suggested change.

*Line 57: replace "according to" with "as inferred from U/Th dating and lamina counting between the radiometric ages (: : :.).*

We will rephrase this as suggested. In addition, in response to the comment by Reviewer 2 we will add a bit more detail about the dating methods we used to arrive at our age model for this study.

*Line 65: there are many papers before the Regattieri et al. (2016) which demonstrated the importance of trace elements in tracing the seasonality of hydroclimate in stalagmites. Please refer to Treble et al., 2003; Fairchild and Treble (2009), Griffiths et al., (2010).*

We thank the reviewer for the suggestion and will update our references with the suggested papers.

*Line 70: actually, the first to use trace elements to fingerprint volcanic events were Frisia et al. (2005). The most accurate expression here would be: and can be used to identify changes in atmospheric load of anthropogenically and volcanic derived aerosols, as well as volcanic ash*

We will rephrase this sentence to clarify and refer to Frisia et al., 2005.

*Line 71: please add after geochemical layering : "of Proserpina"*

We will add this to the sentence.

*line 73: replace "the relation" with its relation. Replace "this work" with the present study.*

We will make the suggested changes in the revised manuscript.

*Line 75: I would rephrase for clarity: To achieve: : :.three sections of the Proserpina stalagmite, each encompassing 12 years of growth, were analysed, namely : P16 (from: : :.to), P17 (from to ) and P 19 (from : : :to). Each section was analysed at sub-annual resolution : : :..  this sentence is not entirely clear. Do you mean: the results from the selected sections were then applied to interpret in terms of seasonality of hydroclimate variability climate variability the Proserpina stable isotope ratio curves for the last 400 years. Please, check if I understood correctly what was meant with your sentence. I looked at the figures : : : but I may be wrong.*

We understand the confusion caused by this sentence and thank the reviewer for pointing this out. We will rephrase to clarify that we will interpret the chemical changes in the three intervals in the context of the longer stable isotope record that is available from Van Rampelbergh et al. (2015) to arrive at a discussion of climatic changes over the past 400 years. Please note that the most recent interval ('P19' now to be renamed to 'P20') is longer than 12-years and we will mention this specifically to avoid confusion.

*Line 78 replace elemental with element*

We will correct this typo.

*Apologies, I have become a bit annoyed at having to correct grammar and typos: : :I have started highlighting issues in yellow on the pdf to save my time.*

We understand the choice of Reviewer 3 to annotate typographic comments on the PDF to save time and regret that this was necessary. We appreciate the time and effort the reviewer spent to make these corrections and will go through her/his annotations in detail and amend them in our revised manuscript.

*Line 87 to 91 : local is too vague. What does it mean? Partially dolomitized limestone? As the geology is relevant to the data and discussion, a geologic section of the beds above the cave and inferred distribution of dolomite should be provided. Fig. 1 does not give any geologic information. Somehow, the text is referring to formations that are not shown, the relevance of which is, thus, unknown to the reader. In the opinion of this reviewer, this is not scientifically acceptable, as it fails to provide crucial information.*

This is a valid comment and we will make an effort to update Figure 1 to contain information about the local geology of the region. With "local" in this context we refer to the geology directly overlying the Han-sur-Lesse cave system. This geology is more or less consistent throughout the region and is part of the larger synclinorium of Dinant which is a structure of folded (mostly) Devonian limestones that covers most of the Ardennes region in southern Belgium. We will limit our description of the geology to the direct ("local") area of the cave system for sake of brevity:

Specifically, the Salle du Dôme opens in the Devonian Givetian limestone beds forming an anticline structure, which explains the surface geomorphology of the hill in which the cave is located (Delvaux de Fenffe, 1985). The limestone reaches a thickness of 20-50 m above the cave, as estimated by the map of the cave and the surface. Some of the Devonian beds are dolomitized. Since no impermeable formations are present above the cave, precipitation directly seeps through the thin (~25 cm) soil and enters the epikarst.

*Lines 95-97: Just a suggestion – it would be great help to the reader if a figure with distribution of rainfall and temperatures throughout the year (or a 30 year mean) were presented. True, we are referred to Genty &Deflandre (1998) and Genty & Quinif (1996), but still it is beneficial to have it visually in the present manuscript, even as Supporting Material. It would help understanding the significance of proxy data.*

This comment was also made by Reviewer 1. The necessary information is currently present in Figure 8, but we will move this figure forward in the manuscript to refer to it earlier and give the reader an overview of the present-day (1999-2012 average) climate in the region.

*Line 103: I suggest to write that it was removed in 2001.*

We agree that this is information we should add and will do so in the revised version.

*Line 108: colder than what?*

Colder than the air temperature in the cave mentioned in the previous sentence. We will add this for clarity.

*Line 109: what is the time scale of "temporary"?*

See Figure 8 (to which we will refer in the revised version): This increase in rainfall in summer lasts for approximately 2 months (July-August). We will mention this in the revised version.

*Line 114 replace "record" with values.*

We will rephrase this.

*Line 117: what does tabular mean? Better to rephrase: flat topped surface. Even if tabular was used by Rampelbergh et al. (climate of the past 2015), still the speleothem looks more a composite edifice showing, clearly, from the insert in Fig 1. From Fig. 1 (and Fig. 1b in Rampelbergh et al., 2015) Proserpina it is not a typical stalagmite, but a speleothem that formed through the coalescence of several stalagmites, because of the fast drip rate, into a stalagmitic flowstone characterized by mm- to cm- scale rimmed pools (which is also clear from Verheyden et al., 2006, Figures 1 and 2 showing cahaotic "pool*

*like" deposits). Which hints to the fact that the feeding drip points may be multiple, shift in time and drip rates may have been extremely variable because of rupture of feeding straws (Verheiden et al, Fig 1). And, in fact, the sentence continues by stating that it is fed by a drip "flow". Thus, I would call it a stalagmitic flowstone.*

Agreed, we will adapt the terminology accordingly to be more precise. We would like to note that the Proserpine did not form through the coalescence of multiple stalagmites. The type of speleothem is described in the French literature as a "tam-tam stalagmite".

*Line 122. State entirely what DCL and WPL are the first time. In any case.*

We will write out "Dark Compact Layers" and "White Porous Layers" here.

*Line 156: The way the sentence is written, it would seem that there 50 years of uncertainty > I suppose that "at" should be replaced by from 1960 (to 2000).*

We will rephrase this for clarity, stating that layer counting is accurate between years 1960 and 2001.

*Line 231 It is unclear to this reviewer why line P17 was taken at an angle with crystal growth. In Fig. 2 B one can see that the directions of columnar crystal growth at the core of the photo is from bottom to top. The P17 was taken laterally. It is unclear, from the photo, if it is meant to be perpendicular to an outer layer , roughly at 45 degrees from the "page vertical direction). Yet, from the "blurred" image I could retrieve, it seems that the fabric is more complex, there could be more "mosaic-type" crystals as in P 19. Please explain the direction of P17, which may actually explain the isotope ratios data in Fig 2, specifically the abrupt shifts. I suggest the Authors to provide a better fabric control for their data. His may also be the reason why the thickness are different for P16 and 17 relative to 19. In P 19 it is clear that the transect cross cuts laminae perpendicularly.*

All three transects used in this study were sampled perpendicular to the prevailing local direction of the annual layering. In the case of piece P17, this direction is indeed at an angle of ~25° with respect to the vertical in the picture. This direction is chosen because the layers at this location in the speleothem are roughly parallel and at an angle of 25 degrees with the horizontal in the picture. To emphasize this, we will accentuate the layers (which we concur are hard to spot on the current image) in all three pieces in Figure 2 to show that the transect is indeed chosen perpendicular to the layering.

*Line 234. Perhaps there is an "of" missing?*

We will add "of" after "effect".

*Line 288. Perhaps better to expand for those who are not familiar. Rather than "same samples" elaborate: whilst stable isotope ratios are measured on powders drilled: : : trace elements are: : :thus the exact location of the analyses in the same sample may not coincide: : :.*

We will rephrase this according to the reviewer's suggestion and agree that this would be clearer.

*Line 297. What is intended here for mineral? Dissolved ions? And what is intended for organic matter? Please explain. Have any analyses for the nature of the DOC in the H sur L waters been carried out? The type of Organics? LMW-A, LMW-N, building blocks etc? Works by Hartland et al. (2012), Rutlidge et al. (2014) are showing a connection between the NOM and trace element transport in cave waters (and trace element content in stalagmites). Thus, in order to better understand the trace element variability, a knowledge of DOC in H sur L waters is needed (see the speculation at Line 8. This can be solved by having a characterization of NOM in the feeding waters).*

We did not conduct a separate analysis of the nature of OM in the drip water, but we have information about conductivity (ion concentrations; Genty and Deflandre, 1998) and variability in Mg/Ca and Sr/Ca of dripwaters (Verheyden et al., 2008), which are higher during higher drip rates in autumn after the more intense July-August rainfall. At the same time, Verheyden et al. (2008) postulate that autumn flushing brings humic and fulvic acids that accumulate due to intense biological activity above the cave during spring and summer. This lag in response is also in agreement with the results of cave monitoring done by Van Rampelbergh et al. (2014) which shows a peak in discharge in October-November as a delayed response to summer rainfall. This seasonality explains much of the observed trace element and stable isotope variability as well as the presence of the layering in the speleothem mineralogy. We will try in the revised version to explain this more clearly earlier in the manuscript to help the reader to better understand the cave system and refer to it in our discussion.

*Line 316 to 323. The discussion about P becomes highly speculative and citing Frisia et al (2012) is, here, not the case, as the case-study in the cited paper is from locations where P is known to be present in the host rock and/or is related to stromatolite-like features. Unless the Authors show petrographic features that resemble microbially induced ones, the hypothesis has no grounds. And the same applies for the host rock, Authors do not provide a geochemical characterization of the rock. Hence, Authors simply do not know where P may come from and why it behaves the way it does. This is why they need to provide a DOC characterization of the dripwater. As there are organic acids that favour the mobilization of P. Each cave has its own environment and diverse types of soil. Also: Vegetation dieback for P was hypothesized in a seminal paper by Treble et al (2003), which I believe should be cited, as it provides a diverse context that that of Baldini et al (2002).*

In the revised version, in response to the comment from multiple reviewers, we will provide records for P in Figure 5. However, P does not show the same seasonality that is so clearly expressed in the other trace elements. Therefore, even adding this record to Figure 5 will not help is explain why P is behaving the way it does. If P was predominantly mobilized by humic and fulvic acids, one would expect a seasonal pattern that follows the autumn increase in discharge, in which these acids are supposedly enriched. We do not observe this however, so the P record remains hard to discuss (see also comment above). We will mention the studies suggested by the reviewer, but will likely not venture beyond highlighting the different processes that might influence the P concentration in the Proserpine speleothem as we are not certain about which processes are dominant.

*Lines 339 to 393 The issue here is that the different behaviour from P19, P16 and P17 is that P19, from Fig 2, is the only clear transect that is perpendicular to parallel growth layers that are perfectly flat. The delta 13C values and flat tops with a seemingly predominant compact fabric indicate that P19 formed from a thin film of fluid that resided for a longer time (than P17 and 16) at the top surface of the growing stalagmite. This allowed prolonged degassing. At this stage, the correlation between delta 13 C, Mg and Sr may be slightly different than just PCP, the: : :.delta 13 C may be also influenced by in- cave degassing, but all processes lead to the same conclusion: less infiltration. For the antiphase correlation between Mg and Sr, the Authors speculate about a role for dolomite dissolution, albeit the dolomite bodies are not presented in a geologic section. If this were the case, it should hold also forP19, unless the hydrological pathway changed completely. But there is another explanation, for which the Authors should consult Rutlidge et al (2014). The Authors should actually analyse other elements, such as Al, K, Si and check their behaviour in relation to Mg and Sr. Works that use Sr isotopes have highlighted that Sr may have a twofold provenance: from soil and from rock (Belli et al.,2017). Mg has also a similar twofold provenance (Rutlidge et al., 2014). Now, if Sr had provenance from dolomite and limestone, the Authors should explain why the overall Sr values in P19, 16 and 17 are similar. If Sr in P19 came only from limestone, why is it that the baseline is the same (about 60 ppm?). Also, the Mg concentration in P 19 is overall higher than in P16 and 17, which should not be expected if the hydrological pathway was in a pure limestone, or dolomitized limestone. So, the actual possibility is that something changed between 16 and*

*17, rather than "somewhat" between P16 and P19., but has nothing to do with dolomite (or little to do with it). It is the Y and Zn cycles that actually point to that. It could be soil. The other problem is that the Author cite shales. And shales may be rich in Sr. And Zn and Y Thus, trace elements and/or Sr isotopes that point to provenance of Sr (soil or shales) would give robustness to the speculation.*

We appreciate the discussion by Reviewer 3 and will try to incorporate as many elements as possible from it into our discussion of the phase relationship between trace elements and stable isotopes. The cross plot between these proxies which was suggested by Reviewer 2 might also help here, and we will add it to our revised manuscript. We agree that a change in land use might be an aspect we currently overlooked in our discussion and we will add it in the new version. As we will try to show in an updated version of our Figure 2, all transects are in fact taken perpendicular to the speleothem layering. Nevertheless, we will acknowledge that a change in the morphology of the speleothem itself, resulting in prolonged degassing, might be part of the explanation for the correlation between d13C and Mg and Sr concentrations. Finally, the previously suggested addition of information about the overlying geology should clarify whether or not dolomite or shale dissolution contributed to the trace element budget in a significant way. We will update our discussion in this section taking this new geological information into account. Evidence for the fact that the overlying rock is partly dolomitized comes from a characterization of the rock in Verheyden et al. (2000), which we will cite in this section. We will also add more literature references to discuss the antiphase correlation between Mg and Sr (e.g. Pingitore, 1978; Pingitore and Eastman, 1986; Pingitore et al., 1992; Van Beynen et al., 1997). In addition, we will list the different processes that could cause the observed anticorrelation between Mg and Sr, as suggested by the reviewer and by our reply above. In this discussion, we will acknowledge that incongruent dissolution is one of the possible hypothesis.

**Pingitore N.E., 1978**. The behavior of $Zn^{2+}$ and $Mn^{2+}$ during carbonate diagenesis: theory and application. Journal of Sedimentary Petrology 48 (3): 799-814.
**Pingitore Jr. N.E. and Eastman M.P., 1986.** The coprecipitation of Sr2+ with calcite at 25ºC and 1 atmosphere. Geochimica et Cosmochimica Acta, 50: 2195-2203.
**Pingitore N.E., Lytle F.W., Davies B.M., eastman M.P., Eller P.G. and Larson E.M., 1992.** Mode of incorporation of Sr2+ in calcite: determination by X-ray absorption spectroscopy. Geochimica et Cosmochimica Acta 56: 1531-1538.
**Huang Y., Fairchild I.J., Borsato A., Frisia S., Cassidy N.J., McDermott F., Hawkesworth C.J., 2001.** Seasonal variations in Sr, Mg and P in modern speleothems (Grotta di Ernesto, Italy). Chemical Geology 175: 429-448.
**Van Beynen P.E., Toth V.A., Ford D.C. and Schwarcz H.P., 1997.** Seasonal fluxes of humic substances in cave drip waters, Marengo Cave, Southern Indiana. Proc. of the 12th Int. Congress of Speleology, Switzerland (2): 120.

*Line 415Incongruent dolomite dissolution is not really demonstrated by the data, as I mentioned above, there should be a change in the mean Mg and Sr values from P19 and P17/16, there is not. The baseline is similar. Also, in P17 and 16 the Mg is not always co-varying with the delta13C (Fig.5). If IDD were acting, I would expect that higher Mg coincided with more positive delta 13C, because of dolomite dissolution. But when the Mg is high and delta 13C is more negative, that there is an issue. Unless Mg comes from the soil, or the dolomitized shales have a low delta13C because they are rich in organic matter (I suppose they are dark, but I may be mistaken). To support the IDD hypothesis, I sugget that the Authors provide a geochemical composition for the limestones, the shales and the dolomite, which would help understanding the effects of the processes they invoke (PCO vs IDD)*

As mentioned above, we will add this geological information earlier in the manuscript and refer to it in this part of the discussion to investigate the possibility of IDD or shale contribution playing a role. The hypothesis explaining the lack of covariance between Mg and $\delta^{13}$C that does include IDD would be that if the water is saturated with respect to CaCO3, calcium carbonate starts to precipitate in the epikarst, leading to increased degassing and an increase in $\delta^{13}$C. The changes in $\delta^{13}$C are therefore mostly related to degassing, not IDD, and the relationship between δ¹³C and Mg is not necessarily always positive, as they are governed by independent processes. We will explain this more clearly in our discussion and mention that the dominance of these processes depends on the composition of the dolomites and limestones in the epikarst, as suggested by the reviewer.

*Line 448 I do not understand why a change in "cave morphology" is cited here. IWhat would be important, from the data presented by the Author, is a change in ENVIRONMENTAL C6 parameters. Not necessarily climate. This reviewer admits a poor knowledge of the studied region, but being in Belgium, I suppose that humans have somewhat impacted the land. Even if it does not appear "visible", land use (tree cutting, particularly in the LIA, when people had to find wood somewhere: : :.) may have altered the hydrological pathways, changed soil (uprooting). These options: : :of anthropogenic disturbance, need to be investigated also for P 17 and 16, not just P19.*

This is a valid point by Reviewer 3, and we will broaden our discussion of "climatic change" to encompass "environmental change" which may also entail changes in land use as an explanation for the observed variations. In an earlier stage of working on this study, we tried to find data on the land use change in this area in the 17th century, but sadly such information is not available. We can therefore not venture beyond speculation about changes in land use. We will cite the discussion of land use change in the literature (e.g. Van Rampelbergh et al., 2015). Forest cover was reduced between the little ice age and the early 1900's, but mostly on the more humid slopes rather than the drier tops of the hills. Since the Salle du Dôme is situated under the top of the hill (formed by the anticline structure mentioned above), changes in local forest cover are relatively small, but cannot be neglected.

*Line 427: may also indicate that soil disturbance/forest disturbance above the cave resulted in faster infiltration for P19, because of less soil humidity retention. And, obviously, the rise in temperature in the last 150 years related to the end of LIA, industrial "revolution" and increase in GHG in the atmosphere.*

Agreed, we will add this possibility in our discussion and provide the proper citations.

*Line 473-489 This part of the discussion does not account for land use. Because the data are discussed with the assumption of PCP versus IDD. But this reviewer fails to see unequivocal proof of IDD from the trace element data.*

See above, we will adapt our discussion to include the possibility of a change in land use explaining (part of) the changes in speleothem chemistry we observe in this study. In general, as mentioned above, we will more clearly list the different possible hypotheses and the evidence that could point towards them.

*Lines 490-517 Actually, the best piece of information in favour of higher recharge in the 17th century is the lamina thickness, given that the delta13C values seem similar over the three periods. The delta 18O could reflect cooler air masses in the 17th century, or a diverse provenance. So it may not be an unequivocal marker of enhanced rainfall in winter. But the thicker laminae in a cold period need to have water supply and an efficient soil/microbes/fungal hyphae. The marked seasonality could well be a marker of snowmelt, which should be considered as hypothesis. Actually, if the bulk of infiltration in P16 and 17 came at snowmelt rather than in Autumn (as I suppose there is no historical record of rainfall patterns for the 17th Century), then the behaviour of P may be explained: : :there is no vegetation dieback in Spring.*

We thank the reviewer for this suggestion and will try to implement it in our discussion in the revised version of our manuscript (albeit probably earlier on in the discussion where the P record is discussed). This comment is similar to a comment made by Reviewer 2, namely that the difference in temperature might have driven the difference in d18O between the 17th century and the present day. However, this change is not entirely clear from the longer d18O record, so we suggest that temperature is not the dominant factor. Rather, a change in the d18O of precipitation (more snowfall could indeed contribute to this) may explain the difference. More specifically, the different proposed processes might be linked, such that an change in land use causes a decrease in vegetation cover which affects infiltration of precipitation water. A change in the precipitation and evapotranspiration pattern can contribute to this effect. Finally, snow melt may have the so-called "piston effect" pushing older water out of the epikarst and therefore increasing the flow of water to the speleothem site, causing increased seasonality in Mg and Sr. We will add this explanation which links anthropogenic and climate and environmental changes in our discussion.

*Hope these comments help.*

*Please also note the supplement to this comment:*

*https://www.clim-past-discuss.net/cp-2019-78/cp-2019-78-RC3-supplement.pdf*

Kind regards on behalf of all the authors,

Niels de Winter

**Reconstructing seasonality through stable isotope and trace element analysis of the Proserpine**

**stalagmite, Han-sur-Lesse Cave, Belgium: indications for climate-driven changes during the last 400**

**years**

Stef Vansteenberge[1*], Niels J. de Winter[1], Matthias Sinnesael[1], Sophie Verheyden[2,1], Steven Goderis[1,3],

Stijn J. M. Van Malderen[3], Frank Vanhaecke[3] and Philippe Claeys[1].

[1]Department of Analytical, Environmental and Geochemistry, Vrije Universiteit Brussel, Pleinlaan 2, B-

1050 Brussels, Belgium.

[2]Royal Belgian Institute of Natural Sciences, Jennerstraat 13, B- 1000 Brussels, Belgium

[3]Department of Analytical Chemistry, Ghent University, Campus Sterre, Krijgslaan 281 S12, B-9000

Ghent, Belgium

**\*Corresponding author: Niels.de.winter@vub.be**

**Abstract**

Fast growing speleothems allow the reconstruction of palaeoclimate down to a seasonal scale. Additionally, annual lamination in some of these speleothems yields highly accurate age models for these palaeoclimate records, making these speleothems valuable archives for terrestrial climate. In this study, an annually laminated stalagmite from the Han-sur-Lesse Cave (Belgium) is used to study the expression of the seasonal cycle in northwestern Europe during the Little

Ice Age. More specifically, two historical 12-year-long growth periods (ca. 1593-1605 CE and 1635-

1646 CE) and one modern growth period (1960-2010 CE) are  analyzed on a sub-annual scale for their stable isotope ratios ($\delta^{13}C$ and $\delta^{18}O$) and trace element (Mg, Sr, Ba, Zn, Y, Pb, U) content.

Seasonal variability in these proxies is confirmed with frequency analysis. Zn, Y and Pb show distinct annual peaks in all three investigated periods related to annual flushing of the soil during winter. A

strong seasonal in phase relationship between Mg, Sr and Ba in the modern growth period reflects a substantial influence of enhanced prior calcite precipitation (PCP). In particular, PCP occurs during summers when recharge of the epikarst is low. This is also evidenced by earlier observations of increased $\delta^{13}C$ values during summer. In the 17th century intervals, there is a distinct antiphase relationship between Mg, Sr and Ba, suggesting that processes other than PCP, i.e. varying degrees of incongruent dissolution of dolomite, eventually related to changes in soil activity and/or land use change are more dominant. The processes controlling seasonal variations in Mg, Sr and Ba in the speleothem appear to change between the 17th century and

1960-2010 CE. The Zn, Y, Pb and U concentration profiles, stable isotope ratios and morphology of the speleothem laminae all point towards increased seasonal amplitude in cave hydrology. Higher seasonal peaks in soil-derived elements (e.g. Zn and Y) and lower concentrations of host-rock derived elements (e.g. Mg, Sr, Ba) point towards lower residence times in the epikarst and higher flushing rates during the 17th century. These observations reflect an increase in water excess above the cave and recharge of the epikarst, due to a combination of lower summer temperatures and increased winter precipitation during the 17th century. This study indicates that the transfer function controlling Mg, Sr and Ba seasonal variability varies over time. Which process is dominant, either PCP. Soil activity or dolomite dissolution, is clearly climate-driven and can itself be used as a  paleoenvironment proxy.

**Keywords:** Speleothem, seasonality, Little Ice Age, trace element concentrations, stable isotope ratios, proxy transfer functions

**1. Introduction**

Speleothems have been successfully used to reconstruct paleoclimate on various time scales (Fairchild and Baker, 2012), from tropical latitudes (e.g. Wang et al., 2001) to temperate areas (e.g. Genty et al.,

2003). Fast growing speleothems enables paleoclimate reconstructions to reach seasonal resolution, or even higher (Van Rampelbergh et al.,

2014). Some speleothems are characterized by distinct physical and/or geochemical layering, which improves chronologies and lends more confidence to the interpretation of proxy records at these high temporal resolutions (e.g. Mattey et al., 2008; Boch & Spötl, 2008). The occurrence of visible annual laminae in speleothems has been reported from sites all over the world (Baker et al., 2008). A common expression of this visible layering is an alternation of dark compact laminae (DCL) and white porous laminae (WPL), as defined by Genty and Quinif (1996). According to Dreybrodt (1999) and Baker et al.

(2008), the origin of visible seasonal layering is related to seasonal variations in drip rate and in drip water supersaturation.

Additionally, seasonal changes in cave ventilation can influence the intensity of the degassing process of the dripwater and influence the pH

of the fluid from which the speleothem precipitates, producing seasonal variations in crystal fabric (e.g.

Frisia et al., 2003). These two processes can work in conjunction, with drip rate influencing annual laminae thickness while degassing influences the speleothem fabric (Boch et al., 2010). Such changes in drip rate often coincide with the presence of a varying degree of prior calcite precipitation (PCP).

PCP is the process of calcite precipitation upstream of the site of speleothem deposition (Fairchild et al., 2000). An increase in PCP occurs when the ability of infiltration waters to degas increases.

Therefore, a higher degree of PCP is attributed to drier periods (Fairchild et al., 2000; Fairchild and Treble, 2009). Variations in the amount of PCP have been observed on a seasonal scale (e.g. Johnson et al., 2006).

The presence of seasonally laminated speleothems in Belgian cave systems has been known for several decades (e.g. Genty and Quinif, 1996). The best known example is the Proserpine stalagmite, which was  cored  in the cave of Han-sur-Lesse  and first  studied by Verheyden et al. (2006). The speleothem has a well-expressed visual and geochemical seasonal layering over the last 500 years as inferred from and U/Th dating and lamina counting between the radiometric ages (Van Rampelbergh et al., 2015). This geochemical layering is reflected by sub-annual variations of stable isotope ratios ($\delta^{13}$C and $\delta^{18}$O). A thorough understanding of modern seasonal control on variations in $\delta^{13}$C and $\delta^{18}$O in speleothem calcium carbonate results from rigorous monitoring of the conditions at the sample site in Han-sur-Lesse cave as carried out by Van Rampelbergh et al. (2014) for the period 2012-2014.

In addition to the commonly used speleothem $\delta^{18}$O and $\delta^{13}$C proxies, the use of trace elemental concentrations (e.g. Mg, Sr, Ba, Zn and U) as palaeoclimate and palaeoenvironmental proxies is becoming standard practice in speleothem reconstructions (Fairchild et al., 2000; Treble et al., 2003; Fairchild and Treble, 2009; Griffiths et al., 2010; Regattieri et al., 2016). The use of trace elements brings additional information that can be used to unravel seasonal variability in speleothem chemistry. Examples of this include the use of trace element concentrations as proxies for precipitation (Baldini et al., 2002; Warken et al., 2018), soil processes (Regattieri et al., 2016) or changes in sediment supply (Regattieri et al., 2016).  They have also been linked to changes in the atmospheric load of anthropogenic and volcanic derived aerosols, as well as volcanic ash fall events from speleothem records (Frisia et al., 2005; Jamieson et al., 2015).

The first objective of this study is to better characterize the geochemical layering of the Proserpine speleothem by adding trace element proxies to improve the understanding of processes driving the geochemical layering and to further resolve  its relation with seasonal climatic variability. In addition, the present study  compares the seasonal cycle within earlier identified cold periods (Verheyden et al., 2006; Van Rampelbergh et al., 2015; Supp. Mat. **Fig. 1**) to present-day seasonal signals. To achieve this, two 12-year long stalagmite growth periods (1593-1605 CE ± 30, hereafter P16 and 1635-1646 CE ± 30, hereafter P17) and a more recent  growth period deposited between 1960-2010 CE (hereafter referred to as P20) were analysed at a sub-annual scale for their stable isotopic ($\delta^{13}$C and $\delta^{18}$O) and trace elemental variations. The names of these growth periods (roughly) refer to the century in which they were deposited (16[th], 17[th] and 20[th] century respectively). Chemical changes in these three intervals will be interpreted in the context of the longer
stable isotope record that is available from Van Rampelbergh et al. (2015) to discuss changes in
hydroclimate and seasonality over the past 500 years as recorded in the Proserpine speleothem.

**2. Geological setting**

**2.1 Han-sur-Lesse Cave**

With a total length of approximately 10 km, the Han-sur-Lesse Cave system, located within a limestone
belt of Middle Devonian age, is the largest known subterranean karst network in Belgium (**Fig. 1A**). The
cave system was formed by a meander cut-off of the Lesse River within the Massif de Boine, which is
part of an anticline structure consisting of Middle to Late Givetian reefal limestones (i.e. the Mont-
d'Haurs and Fromelennes Formations (Fm.); Delvaux de Fenffe, 1985; **Fig. 1C**). The limestone epikarst
reaches a thickness of 20-50 m above the cave, as estimated by the map of the cave and the surface.
(Quinif, 1988).
Studies have shown the  presence of dolomite in these Givetian limestones which are directly
overlying the Han-sur-Lesse cave (**Fig. 1**). Within the Mont-d'Haurs Fm., the biostromal limestones are
alternated with fine-grained micritic limestones and dolomitic shales (Preat et al., 2006).
Additionally,  studies by Verheyden et al. (2000) and Pas et al. (2016) on the Middle
Devonian  strata in which the Han-sur-Lesse cave is situated  have shown that dolomitized
beds also occur within the limestones of the Fromelennes Fm. Specifically, the Salle du Dôme, in which
the Proserpine is located, opens in the Devonian Givetian limestone beds forming an anticline
structure, which explains the surface geomorphology of the hill in which the cave is located (Delvaux
de Fenffe, 1985; **Fig. 1**). Since no impermeable formations are present above the cave, precipitation
directly seeps through the thin (~25 cm) soil and enters the epikarst.

The Han-sur-Lesse Cave is located ~200 km inland at an elevation of 200 m above sea level. The region
is marked by a warm temperate, fully humid climate with cool summers, following the Köppen-Geiger
classification (Kottek et al., 2006). In the period 1999-2012, the region
experienced a seasonality in monthly temperatures between 2.5°C (January) and 18°C (July), with a
mean annual temperature of 10.2 °C (**Fig. 2**). Precipitation is fairly constant year round
around 40-50 mm month$^{-1}$, with two months (July and August) experiencing increased precipitation of
90-95 mm month$^{-1}$. Mean annual precipitation was 820 mm yr$^{-1}$ in
Rochefort, 10 km from Han-sur-Lesse (Royal Belgian Meteorological Institute, Brussels, Belgium). The
study site is affected by a North Atlantic moisture source all year round (Gimeno et al., 2010) and the
amount of precipitation does not follow a seasonal distribution. Calculations applying the

Thornthwaite formula (Thornthwaite and Mater, 1957) show that there is a strong seasonal trend in the water excess, i.e. the amount of rainfall minus the amount lost by evapotranspiration, with water excess only occurring from October to April (Genty and Deflandre, 1998; Genty and Quinif, 1996).

The studied speleothem was retrieved from the Salle-du-Dôme in 2001, a 150 m wide and 60 m high chamber that formed by roof-collapse of the limestone (**Fig 1B**). The Salle-du-Dôme is well ventilated, as it is located close to the cave exit and connected through two passages to nearby chambers.

Monitoring of cave atmosphere within the Salle-du-Dôme for the period 2012-2014 showed that in

2013 the temperature inside the chamber varied seasonally between 10.5 and 14.5 °C (Van

Rampelbergh et al., 2014). Similar seasonal trends in temperature are observed for the drip water sampled at the site of the Proserpine speleothem, but the averagemean is 0.5 °C colder than the outside air temperature. The $pCO_2$ of the cave air averages around 500fluctuates between 400 ppmv and 1000 ppmv (in July and August), and averages 500 ppmV for during the whole year. Yet, iIn summer (July-August), a rapid and temporary (2 month) increase to 1000 ppmv ppmV is observed. Also during summer, rainwater $\delta^{18}O$ and $\delta D$ above the cave increase by 3 ‰ and 30 ‰ (VSMOW, Vienna Standard

Mean Ocean Water) respectively, likely due to the atmospheric temperature effect as described by

Rozanski et al. (1992). In contrast, drip water $\delta^{18}O$ and $\delta D$ remain fairly stable throughout the year, with averagemeans of -7.65 ‰ and -50.1 ‰ VSMOW and standard deviations of 0.07 ‰ and 0.6 ‰

VSMOW, respectively. During late summer (September), an increase of 1.5 ‰ is observed in the $\delta^{13}C$

record values of dissolved inorganic carbon (DIC) within the drip water

**2.2 Proserpine speleothem**

The Proserpine speleothem is a 2 m high, tabular shapedstalagmitic flowstone stalagmitewith a flat top. The speleothem has a surface area of 1.77 m² and is fed by a drip flow with drip rates ranging between 100 and 300 mL min$^{-1}$. The type of speleothem is described in the literature as a "tam-tam speleothem" and is characterized by the occurrence of millimetre to centimetre scale rimmed pools, which causes chaotic "pool-like" deposits to occur in parts of the speleothem (Verheyden et al., 2006).

The speleothem grew over a period of approximately 2 kyr and has thus has an exceptionally high averagemean growth rate of 1 mm yr$^{-1}$. The This large speleothem was drilled and a 2 m long core was retrieved. The upper 50 cm of this core, dating back to approximately 1500 CE (Supp. Mat. Fig. 2), shows a well-expressed layering of alternating Dark Compact Layers (DCL) and White Porous Layers (WPL (; Verheyden et al., 2006). Previous studies concluded that simultaneous multi-decadal simultaneous changes in different proxies (such as crystal fabric, growth rate, layer thickness, and oxygen and carbon stable isotope ratios) indicate that these are controlled by common climatic, environmental or anthropogenic factors, despite the observationand that some parts of the Proserpine speleothem  have been deposited out of isotopic equilibrium with the drip water (Verheyden
et al., 2006; Van Rampelbergh et al., 2015).

Based on a detailed cave monitoring study at the Proserpine site in the years 2012 to 2014, Van
Rampelbergh et al. (2014) showed that $\delta^{18}O$ and $\delta^{13}C$ of seasonally deposited calcite reflect isotopic
equilibrium conditions and that variations of stable isotope ratios are induced by seasonal changes.
These seasonal changes in stable isotope ratios correspond with the observed visible layering. The
speleothem $\delta^{18}O$ value is believed to reflect changes in seasonal cave climatology. While drip water
$\delta^{18}O$ remains constant, calcite $\delta^{18}O$ decreases  by ~0.6 ‰ in summer months, caused by
temperature-dependent fractionation during calcite precipitation. This fractionation was calculated to
be -0.2 ‰ °C$^{-1}$ (Van Rampelbergh et al., 2014). In contrast, $\delta^{13}C$ reflects seasonal changes occurring at the epikarst level. A ~1.5 ‰ increase of $\delta^{13}C$ in drip water DIC during late summer is directly reflected
in the freshly deposited speleothem calcite. The enrichment in drip water $\delta^{13}C$ values occurs shortly
after the observed decrease in drip water discharge, and therefore seasonal variations in the degree
of prior calcite precipitation in the epikarst has been hypothesized to be the main driver of seasonal
$\delta^{13}C$ changes in the drip water (Van Rampelbergh et al., 2014).

**2.3 Dating**

The age-depth model of the Proserpine speleothem core has been established and discussed by Van
Rampelbergh et al. (2015) and is provided in the supplementary material (Supp. Mat. Fig. 2). This age-
depth model was constructed by using a combined approach of U-Th radiometric dating, based on 20
U-Th ages, and layer counting. It was shown that the  number of counted layers is in
good agreement with the U-Th ages (see **Table 1** in Van Rampelbergh et al., 2015). However,  9 to

10 cm from the top of the core, a perturbation with heavily disturbed calcite occurs, making it
impossible to construct a continuous layer  counting chronology. Remains of straw and soot
were found within this perturbation, suggesting that at that time, fires were lit on the speleothem's
palaeosurface (Verheyden et al., 2006). Layer counting gave an age of 1857 ± 6 CE for the
reestablishment of calcite deposition after the perturbation and U-Th age-depth modeling showed
that the start of the perturbation occurred at 1810 ± 45 CE (Van Rampelbergh et al., 2015).
Radiocarbon dating of the straw fragments embedded in the calcite gave an age between 1760 and
1810 CE, with 95.4 % probability. The age of 1810 ± 45 CE is used to restart the layer counting after the
perturbation towards the bottom of the core. This gave an age of 1593 to 1605 ± 30 CE for the P16 and
1635 to 1646 ± 30 CE for the P17 section. The more recent section P20 studied here is situated
above the perturbation and its age could be confidently established through annual layer counting
between 1960 and 2001 CE. The same chronology for the Proserpine speleothem was previously used in conjunction with petrographic evidence to show that decreases in cave visitation coincided
with the two World Wars, highlighting the accuracy of the age model (Verheyden et al., 2006).

**3. Methods**

**3.1 Analytical procedures**

The three growth periods studied are shown in **Fig. 2 3** and their age is derived from an age-depth
model based on U-Th-dating and layer counting (Verheyden et al., 2006; Van Rampelbergh et al., 2015;
Fig. S5; see **2.3**). The number of counted layer couplets over the annually layered 500-years determined
the seasonal character of the layers and demonstrated that two layers (one DCL and a WPL) were
deposited per year. The number of years obtained by layer counting between two U-Th datings was
compared with the number of years suggested by the U-Th ages. We combine results of both
independent dating methods to produce the final age model. All growth transects are sampled parallel
to the local direction of growth of the Proserpine speleothem and perpendicular to the growth laminae
(see **Fig. 2**).

For $\delta^{13}$C and $\delta^{18}$O analysis, powder samples are were acquired using with a Merchantek Micromill
(Merchantek/Electro Scientific Industries Inc. (ESI), Portland (OR), USA, coupled to a Leica GZ6, Leica
Microsystems GmbH, Wetzlar, Germany) equipped with tungsten carbide dental drills with a drill bit
diameter of 300 μm. The powders are stored in a 50 °C oven prior the analysis to avoid $\delta^{13}$C and $\delta^{18}$O
isotopic contamination. Measurements for P16 and P17 are carried out on a Nu Perspective isotope
ratio mass spectrometer (IRMS) coupled to a Nucarb automated carbonate preparation device (Nu
Instruments, UK) at the Vrije Universiteit Brussel (Belgium). The $\delta^{13}$C and $\delta^{18}$O records of P16 and P17
consist of 201 and 116 data points, respectively, resulting in temporal resolutions of ~20 and ~10 data
points per year, respectively. The analysis of the P19P20 interval is an extension of the previously
published seasonally resolved 1976-1985 transect (Van Rampelbergh et al., 2014) and is was carried
out on a Delta plus XL IRMS coupled to a Kiel III carbonate preparation unit (Thermo Fisher Scientific,
Germany) also at the Vrije Universiteit Brussel. For P19P20, a total of 350 samples are were analysed,
providing a temporal resolution of ~7 data points per year. All results are displayed as ‰VPDB (Vienna
Pee Dee Belemnite) with the individual reproducibility reported as 2 standard deviation (SD)
uncertainties. Within each batch of ten samples, the in-house reference material MAR2-2, prepared
from Marbella limestone and calibrated against NBS-19 (Friedman et al., 1982) is measured together
with the samples to correct for instrumental drift ($\delta^{13}$C: 3.41 ± 0.10 ‰ (2s2 SD) VPDB; $\delta^{18}$O: 0.13 ± 0.20
‰ (2s2 SD) VPDB). All results are displayed as ‰ VPDB (Vienna Pee Dee Belemnite) with the individual

Total uncertainties for $\delta^{13}$C and $\delta^{18}$O are 0.03 ‰ and 0.09 ‰ (1 SD) for the Nu Perspective setup. With the

Delta plus XL setup these are slightly higher, being 0.04 ‰ and 0.10 ‰ for $\delta^{13}$C and $\delta^{18}$O, respectively (Van Rampelbergh et al., 2014).

Trace element variations are determined using inductively coupled plasma-mass spectrometry complemented by a laser ablation sample introduction system (LA-ICP-MS) at Ghent University (Belgium). The LA-ICP-MS setup consists of a 193 nm ArF*excimer Analyte G2 laser ablation system (Teledyne Photon Machines, Bozeman, MT, USA) coupled to a single-collector sector field 'Element XR'

ICP-MS unit (Thermo Fisher Scientific, Bremen, Germany). The laser  was used to sample adjacent positions along a line segment parallel to the  growth axis. The positions  were ablated one-by-one for 15 s with a laser spot size of 50 μm in diameter, a repetition rate of 30 Hz and a beam energy density of 3.51 J cm$^{-2}$. The line segments for P16, P17 and P20 are drilled at 287, 249 and

445 individual positions, respectively. Sampling via individual laser spots  was preferred over the conventional approach of continuous line scanning because the single positions can be sampled longer, resulting in an improved limit of detection. To carry out the analyses, the speleothem sections and reference materials  were mounted in a HELEX 2 double-volume ablation cell. The

Helium carrier gas (0.5 L min$^{-1}$) is mixed with Argon make-up gas (0.9 L min$^{-1}$) downstream of the ablation cell, and introduced into the ICP-MS unit, operated in low mass-resolution mode. Transient signals for  magnesium (Mg),  aluminium (Al),  silicon (Si), phosphorus (P),  sulphur (S),  potassium (K),  iron (Fe),  manganese (Mn),  zinc (Zn),  rubidium (Rb),  strontium (Sr),  yttrium (Y), barium (Ba),  lead (Pb),  thorium (Th), and  uranium (U) were monitored during analysis of the laser-induced aerosol. Cool plasma conditions (800 W RF power) are used to reduce

Argon-based interferences and to increase the sensitivity of the analysis. A gas blank subtraction is performed on the data acquired at each position, based on the signal acquired 10 s prior to the ablation. Precise and accurate trace element concentration data  were obtained from offline calibration, using seven international natural and synthetic glass and carbonate reference materials:

BHVO-2G, BIR-1G, GSD-1G, GSE-1G, and MACS-3 (United States Geological Survey) as well as SRM 610

and 612 (National Institute of Standards and Technology). Ca is used as an internal standard for calibration of the speleothem measurements, following the assumption that the calcium carbonate in the speleothem contains 38 wt. % Ca. Based on the reference materials and settings described, the reproducibility of the produced elemental concentration data  was typically on the order of 5% relative standard deviation (RSD). Limits of detection (LODs) are given in

**Table 2**.

**3.2 Data Processing**

Frequency analysis is applied to study the variations in the different proxy signals, and allows
evaluating which of these proxies fluctuate seasonally. The  benefit of
frequency analysis for assessing seasonal cyclicity in a proxy
in speleothems and other incremental climate archives  was already recognized by
Smith et al. (2009) and de Winter et al. (2017). Furthermore, the method can identify multi-annual
trends or variability at the sub-seasonal level. Frequency analysis is performed using Fast Fourier
Transformations (FFT) of the isotopic and trace element data in the distance domain. The data
were de-trended and padded with zeros. The power spectra are plotted as simple periodograms
with frequencies  in the distance domain (mm$^{-1}$) to allow  interpretation of seasonality
in the data. The significance level (95%)  was evaluated using Monte Carlo noise simulations. The
routine used operates in MATLAB® and  was based on the scripts provided in Muller and MacDonald
(2000; see Bice et al. (2012 for more detail).

An effective method to compare sub-annual variations of different proxies with each other is by
resampling multiple annual cycles at a regular interval and stacking the individual cycles (Treble et al.,
2003; Johnson et al., 2006; Borsato et al., 2007; de Winter et al. 2018). The advantage of this method
is that the phase-relations of the different proxies are preserved (Treble et al., 2003). Annual stacks
are created based on  moving averages to diminish the influence of low-frequency noise on the
annual stacks. The number of points used for moving averages is determined as a function of the
sampling resolution (i.e., 3-point moving average for stable isotope records and 5-point moving
average for trace element records, see **Fig. 5**). Proxy records with well-constrained seasonal variation
are used to define seasonal cycles. In this study, individual years  were selected based on $\delta^{13}$C
(minima) for stable isotope records and Zn (maxima) for the trace element records. Stable isotope
ratios and trace element stacks were created separately (**Fig. 2**). For P16 and P17, all annual cycles
were included in the stack, except for the first and the last one, since there was no guarantee
that these are entirely represented in the record. For P20, only ten years were selected from the
full record to avoid the effect of  multi-decadal variability (see **Fig. 2**) and to maintain an
approach similar to that of P16 and P17. The years are indicated by the red line in **Fig. 5**.

**4.   Results**

The concentration range of each proxy measured in the three different intervals is shown in **Fig. 3**.
For $\delta^{13}$C and $\delta^{18}$O, the  mean values and ranges (minima to maxima) in P20 are significantly
higher than those in P17 and P16. To illustrate the spread in the trace element records, the median was used instead of the mean as  the median is less sensitive to large concentration ranges and outliers. Al, Si, K, Mn, Rb and Th are not included in this study since > 25% of the data falls below the LOD. An exception was made in the case of Y; of which only few data points are retained for P17

(81% of the data is < LOD  and 18% and 36% of the data is < LOD in P20 and P16 respectively.

However, Y data are discussed because of the clear seasonal signal shown in P20 and P16 (Supp.

Mat. Fig. 3 and 5).

Records of stable isotope ratios ($\delta^{13}C$ and $\delta^{18}O$) and trace element (Mg, P, Zn, Sr, Y, Ba, Pb and U)

concentrations are plotted against distance in **Fig. 4**. The occurrence of darker laminae (DCL) in the samples is indicated by blue bands, clearly showing that these annual laminae are thicker in P16 and P17 (mean 1.135 mm and 1.096 mm, respectively) compared to P20 (mean 0.382 mm). For all intervals, the seasonal cycles are well constrained by $\delta^{13}C$, with lower $\delta^{13}C$ values occurring in DCL. Van Rampelbergh et al. (2014) present in their figure 4, the correspondence between stable isotopic compositions of the calcite samples taken by a micromill on a regular spatial sampling interval and a scan of the stalagmite. This correspondence is based on the information available in 2014 and 2015. However, more recent tools combined with the study of a larger portion of the stalagmite demonstrates that contrary to the affirmation of Van Rampelbergh et al, (2014; 2015), lower values in $\delta^{13}C$ are found in the DCL. The mean $\delta^{13}C$ is higher for P20

(-8.36 ‰) compared to P17 and P16 (-9.82 ‰ and -10.04 ‰, respectively). In addition, the amplitude of the individual cycles is larger in P20. Seasonal cycles in $\delta^{18}O$ are much less pronounced. The most distinctive cycles are observed in P20 and some can be identified in parts of P17 and P16 (e.g.

between 4 and 7 mm in P16 or between 3 and 7 mm in P17), while for other parts (e.g. between 7 and

11 mm in P16) they appear to be absent.

Seasonal variations are observed for Mg, Sr and Ba in all three intervals investigated (**Fig. 4**). In P17

and P16, the median concentrations of these elements are similar; 447 and 444 $\mu g\ g^{-1}$

for Mg, 51 and 45 $\mu g\ g^{-1}$ for Sr and 36 and 33 $\mu g\ g^{-1}$ for Ba (**Fig. 5**). However, in P20 concentrations of Mg and Ba are slightly higher compared to the older intervals, i.e. 706 $\mu g\ g^{-1}$ for Mg and 46 $\mu g\ g^{-1}$ for

Ba. This is also the case for Pb and U with concentrations significantly lower in P17 (0.14 and 0.05 $\mu g$

$g^{-1}$, respectively) and P16 (0.14 and 0.07 $\mu g\ g^{-1}$, respectively) and a seasonal cycle that is less pronounced than in P20 (0.37 and 0.18 $\mu g\ g^{-1}$). In contrast, P16 has the highest median concentrations of Zn (54 $\mu g\ g^{-1}$) and Y (0.04 $\mu g\ g^{-1}$) and both elements display a well-defined seasonal covariation. Note that seasonal variability in trace elements in P16 is most pronounced in the first 9

mm, after which the amplitude of variability decreases. The same is not observed in the stable isotope records. Although the concentration of Zn is lower in P20 and P17 (14 and 25 $\mu g\ g^{-1}$, respectively), the seasonal cycle is still present. Similar observations can be made for Y in P20 (0.02 µg g$^{-1}$). Within

P16 and P17, maxima of Zn, Y, Sr and Ba mostly occur within the DCL.

**Figure 4 6** shows an example of the FFT periodograms of $\delta^{13}$C, Mg, Zn and P in P16. Additional periodograms for the other elements in P16, P17 and P20 are included as supplementary data (Supp. Mat. Fig. 3-5). The frequency analysis confirms the clear seasonal cyclicity of $\delta^{13}$C previously observed by Van Rampelbergh et al. (2014 **Fig. 4 6**). The dominant frequency of $\delta^{13}$C in P16 is 0.8

mm$^{-1}$ (**Fig. 4 6**). This corresponds to a period of 1.25 mm, which is in good agreement with an observed mean layer thickness of 1.13 mm (Supp. Mat. Fig. 6). Growth rates based on neighbouring U-

Th dates are slightly lower (0.94 mm/yr on average; see **Table 1**), highlighting the benefit of combining multiple independent dating methods (e.g. layer counting and radiometric dating) in age models to increase the accuracy of dates and growth rate reconstructions. Because of its distinct seasonal cyclicity, the $\delta^{13}$C cycle is used as a reference to deduce whether or not other proxies record the seasonal cycle. Mg and Zn appear to track this seasonal cycle well as their periodograms contain peaks at 0.8 and 0.75 mm$^{-1}$ respectively, corresponding closely to the frequency of $\delta^{13}$C. For Zn, a broader double peak is observed with a main period of 1.18 mm and a smaller period of 1.02 mm. This double peak in the periodogram is caused by small variations in the thickness of the annual cycles around a mean thickness of 1.14 m with a lightly skewed distribution towards thinner layers (see Supp.

Mat. Fig. 6). The P record doesn't display any significant seasonal cycle (95% confidence) (**Fig. 4 4 and**

**Fig. 6**). For P20, visible layers are thinner (mean 0.382 mm) and also the variation in thickness is larger (RSD 28.9%) compared to P16 and P17 (Supp. Mat. Fig. 6). This results in broader and less well defined seasonal peaks in the periodograms.

**5. Discussion**

**5.1 Seasonal cyclicity in stable isotope and trace element records**

A schematic overview of the observed changes in  all proxies  and the interpretation for the three intervals is provided in **Table 2 1**. Assessing the exact phasing of the seasonal cycles of different trace elements to $\delta^{13}$C and the visible layering remains challenging since 1)

A multitude of factors control trace element variations within speleothems and 2) stable isotope ratios and the trace element concentrations are not measured on the same exact sample localitie. Whilst stable isotope ratios were measured on microdrilled powders, trace element concentrations were measured using laser ablation. These records were later carefully aligned based on microscopic observation of sample positions, but due to differences in sample size and sampling density, the exact location of the analyses may not fully coincide. An example of such a phase problem is the occurrence of an additional year in P16 in the trace element curve compared to $\delta^{13}$C (**Fig. 54**, between 1 and 6 mm). Another example is the decrease in amplitude of trace element seasonality which takes place in P16 around 9 mm from the start of the record. This change does not occur in the stable isotope record and therefore complicates the comparison between records in the second part of the record. The fact that all trace element records are effected and not the stable isotope records suggests that this transition highlights a methodological issue, such as a difference in laser beam focus (e.g. due to a difference in polishing quality) during LA-ICP-MS measurements, highlighting the importance of sample preparation for high resolution chemical analyses (e.g. LA-ICP-MS and μXRF).

Nevertheless, overall, $\delta^{13}$C minima generally occur in the DCL, suggesting a similar timing (and maybe control) on the visible laminae and $\delta^{13}$C cycles. Trace element proxies show cyclicity with a similar frequency as the $\delta^{13}$C (**Fig. 46**, ). This in contrast to $\delta^{18}$O, which shows less clear seasonal cycles in P16

and P17 are less clear.

**5.1.1 Zinc, yttrium and lead proxies**

In earlier monitoring studies carried out in the Père-Noël Cave (also part of Han-sur-Less Cave system,

**Fig. 1**), the presence of a late autumn increase in discharge was identified (Genty and Deflandre, 1998;

Verheyden et al., 2008). In-situ conductivity measurements indicated an elevated increase in mineral content (ion concentrations) and/or organic matter increase during this autumnal increase in drip water discharge (Genty and Deflandre, 1998; Verheyden et al., 2008). Measurements of the drip water discharge above the Proserpine stalagmite show that in late October and November, a doubling of the discharge volume occurs. This increased discharge is maintained until May, when a gradual decrease is observed (Van Rampelbergh et al., 2014). In the same period, Mg/Ca and Sr/Ca ratios in drip waters increase as a delayed response to the intense July/August rainfalls (Verheyden et al., 2008). The timing of the elevated discharge agrees with the theoretical water excess occurring above the cave (Genty and Quinif, 1996). At the same time, Verheyden et al. (2008) postulate that autumn flushing brings humic and fulvic acids that accumulate due to intense biological activity above the cave during spring and summer. The observed seasonal cycle in Zn, Y and Pb in the intervals studied is likely caused by this annual autumn-winter flushing. Variations in these trace metal concentrations within annual speleothem layers have previously been attributed to the annual hydrological cycle. For instance,

Borsato et al. (2007) linked the peak concentrations of F, P, Cu, Zn, Br, Y and Pb to the annual increase of soil infiltration during autumnal rainfall. Furthermore, it was suggested that the transport of such elements mainly occurs via natural organic matter (NOM) or other forms of colloidal material.

Enrichments of these soil-derived elements within speleothems are believed to be associated with high drip water flow events (Fairchild and Treble, 2009). Studies have shown that trace metals, such as Cu,

Ni, Zn, Pb, Y and REE, are predominantly transported via complexing complexation by NOM, of which the fraction size in the karstic waters ranges from nominally-dissolved to colloidal-to-particulate
(Hartland et al., 2012; Wynn et al., 2014). In the case of Zn and Pb, Fairchild et al. (2010) have shown
that in Obir Cave (Austria) the visible and ultra-violet lamination forms during autumn and is enriched
in Zn, Pb and P. According to Wynn et al. (2014), the correspondence of distinct Zn and Pb peaks with
the autumnal laminae is compelling evidence for a high-flux transport of these trace metals with NOM.

However, in this study no distinct annual cycle within the P record is observed (**Fig. 4** and **Fig. 6**). The
difference between records of P and other soil-derived elements (e.g. Zn and Y; which exhibit clear
seasonality) is difficult to explain. Phosphorus is considered soil derived as it originates from vegetation
dieback (e.g. Treble et al., 2003; Baldini et al., 2002). Therefore, P has shown similar variations as
observed in Zn, Y and Pb in previous studies (Borsato et al., 2007; Fairchild et al., 2010). In the
Proserpine speleothem, no relation between P and other soil derived trace elements is detected. This
finding is in agreement with minimal seasonal variability observed in δ18O and δD values of cave water
monitored in 2012-2014 (Van Rampelbergh et al., 2014), suggesting that seasonal changes in the
epikarst, linked to water availability, were dominant over seasonal processes related to surface (soil)
processes. If P was predominantly mobilized by humic and fulvic acids, one would expect a seasonal
pattern that follows the autumn increase in discharge, in which these acids are supposedly enriched
(as postulated by Verheyden et al., 2008). However, such a pattern is absent from our data.

An explanation for this can be similar to that proposed by Frisia et al. (2012), being is that P is not
derived from soil leaching, but from other sources such as phosphate minerals present in the epikarst
or microbiological activity. However, no data is available on the P concentrations in the host rock and
no microbially induced petrographic features were observed in the Proserpine speleothem, making it
hard to test this hypothesis.

Alternatively, the lack of seasonality in P might be explained by the occurrence of snowmelt in the
earlier growth periods (P16 and P17). The delay of peak infiltration caused by snowmelt dilutes the
autumn flushing effect and explains the lack of an autumn peak in the P record. Snow melt may have
the so-called "piston effect" pushing older water out of the epikarst and therefore increasing the flow
of water to the speleothem site, causing increased seasonality in Mg and Sr. However, there is no
evidence of this delayed infiltration caused by snow melt in modern cave monitoring (Van
Rampelbergh et al., 2014), while seasonality is also absent in the P record of P20. Therefore, snowmelt
cannot fully explain the trace element patterns observed in the data.

Another alternative explanation might be that the limit of detection of P in our LA-ICP-MS data is higher
relative to the measured values than that of other elements of interest (e.g. Mg, Zn and Sr), causing
higher analytical noise on the P record compared to the other trace element records. The reason might be that P measurements are sensitive to interferences in a Ca-rich matrix (such as calcium carbonate
in speleothems).

[revised manuscript text omitted]

In this studycase, a anmore likely alternative explanation for the P16 antiphase relation in Mg, Sr and

Ba is the incongruent dissolution of dolomite (IDD of CaMg(CO$_3$)$_2$; IDD), taking place during annual periods that are characterized by enhanced water-rock interaction. The presence of dolomite within lateral equivalent Givetian limestone deposits in Belgiumoverlying the cave has been recognised by

Verheyden et al., (2000), Fairchild et al. (2001) and Pas et al. (2016). Dolomitized parts of the limestone host-rock were observed within the nearby Père-Noël Cave (Fairchild et al., 2001). During periods of decreased recharge, i.e. summer for the Han-sur-Lesse Cave, prolonged interaction between water and rock leads to saturation of the karstic water with respect to CaCO$_3$. When saturation is reached, incongruent dissolution of dolomite (IDD) will start and Ca$^{2+}$ concentration remains constant due to the precipitation of calcite (Lohmann, 1988). IDD increases the Mg/Ca of the drip water (Fairchild et
al., 2000), but lowers the Sr/Ca and the Ba/Ca, because dolomite tends to have lower Sr and Ba
contents with respect to calcite (Roberts et al., 1998). During winter recharge, saturation of the water
in the epikarst with respect to calcite is not attained and dolomite does not dissolve. The IDD process
is believed tomay overwhelm the PCP signal in P16 and is likelymight be responsible for the observed
antiphase relation. Since the host rock of the Han-sur-Lesse cave contains both dolomite and shale
deposits (rich in trace elements such as Mg, Sr, Ba and Y), increased host rock interaction of infiltration
waters may influence the trace element concentrations of drip waters (see **Fig. 1**). The dominance of
the interaction with dolomite or shales on the trace element budget depends on the composition of
the host rock and the local pathway of the infiltration water. This makes it very hard to separate these
different processes that influence the trace element composition of the Proserpine speleothem. The
fact that host rock interaction is dominant in P16 but not in P17 and P20 might indicate that in the 16$^{th}$
century, these periods of decreased recharge of the epikarst were more common, suggesting that a
more seasonal precipitation or evapotranspiration regime was in place in the 16$^{th}$ century and that
seasonality in water availability (and therefore epikarst recharge) was reduced in later times. Higher
recharge in the 16$^{th}$ and 17$^{th}$ centuries is also evident from the thicker annual laminae in P16 and P17
compared to P20, showing that the Proserpine speleothem grew faster during these times, which
points towards faster drip rates. During winter recharge, saturation of the water in the epikarst with
respect to calcite is not attained and dolomite does not dissolve.

A third process that could cause changes in the phase relationship between Mg, Sr and Ba is a change
in land use, which changes soil thickness and influences the rate of infiltration of meteoric water into
the epikarst. Both Mg and Sr may have a twofold provenance: from soil and from host rock (Rutlidge
et al., 2014; Belli et al., 2017). A change in land use resulting in a change in the relative contribution of
soil-derived dissolved ions could therefore cause the shifts in phase relationship between Mg, Sr and
Ba observed in **Fig. 7** and **Fig. 8**. Increased seasonal cyclicity in Y and Zn, elements associated with soil
activity, in P16 compared to P17 and P20 actually supports this hypothesis. The soil above the Han-sur-
Lesse cave is very thin (~25 cm), but its thickness might have varied through time. A change in
vegetation cover between the 16$^{th}$ century and later centuries due to, for example, changes in regional
climate or an increase in anthropogenic activity (e.g. forest disturbance) could have brought about
such a change in soil thickness above the cave and explain the changes in trace element patterns
observed in the Proserpine speleothem.

The comparison of the annual stacks for Mg, Sr and Ba of the different intervals corroborates the idea
that PCP is the main process controlling the seasonal variations of these trace elements in P17 and
P19P20 based on the in-phase relation of Mg, Sr and Ba. Within P16, enhanced seasonality in recharge causes IDD or increased concentrations of soil-derived trace elements to dominate over PCP. This is in
agreement with the lower $\delta^{13}$C values for P16, indicating a higher contribution of soil $CO_2$ and explains
explaining the antiphase relation of Mg against Ba and Sr as a consequence of IDD. Somewhere
between the P16 and P19P17 periods, a turnover in the hydrological regime of the epikarst allowed
PCP to become dominant over IDD or soil contribution in the seasonal variations in the proxies. Within
P17, the relationship between Mg, Sr and Ba is less clear. This could point towards a change in
hydrological regime between the periods of deposition of P16 and P19P20, which was still underway
during deposition of the calcite in section P17 (mid-17th century).

**5.1.3 Uranium**

In speleothems, U is thought to be mainly derived from bedrock dissolution (Bourdin et al., 2011;
Jamieson et al., 2016) and to be subsequently transported by the ground water towards the
speleothem (Fairchild and Baker, 2012). The partition coefficient of U is <1 for calcite (Johnson et al.,
2006; Jamieson et al., 2016). This causes U to be preferentially excluded from the calcite and enriched
in the remaining drip water during the process of PCP. However, in P19P20, where PCP is evoked as
the dominant process controlling Mg, Sr and Ba seasonal variations, an antiphase relationship of U
with Mg, Sr and Ba is observed (**Fig. 67**). Johnson et al. (2006) concluded that scavenging of U as uranyl
ion ($UO_2^{2+}$) from the drip water onto the calcite crystal surfaces during PCP has a more dominant
control on seasonal U variability than the partition coefficient. Furthermore, the relatively fast growth
rate of the Proserpine speleothem limits uranyl leaching from the exposed crystal surfaces in the newly
formed speleothem, highlighting the dominance of epikarst processes over those taking place at the
speleothem surface (Drysdale et al., 2019). Such a mechanism may explains why U is antiphase with
the Mg, Sr and Ba variations., especially since the stalagmite is under continuous dripping of water.

**5.2 Seasonal variations in $\delta^{13}$C and $\delta^{18}$O**

To compare and understand the seasonal variations in $\delta^{13}$C and $\delta^{18}$O, annual stacks were created (**Fig.**
**79**) by virtual resampling based on the occurrence of peaks in $\delta^{13}$C values as this proxy reflects the
seasonal cycle best (**Fig. 46**). The minima in $\delta^{13}$C always occur in DCL, for in P16 and P17. In P19P20,
this relationship is less clear, however while on close inspection nearly almostl of the $\delta^{13}$C minima do
also occur within the DCL (**Fig. 54**). Van Rampelbergh et al. (2014) suggested thatobserved seasonal
changes in $\delta^{13}$C of dripwater and of recent calcite, with higher values occurring in summertime when
drip rates are reduced. This led them to conclude that $\delta^{13}$C are is mainly driven by changes in PCP.
Higher $\delta^{13}$C values occur when more PCP is observed, i.e. during periods of lower recharge. The in
phase variations of Mg, Sr and Ba in P19P20 described above supports the hypothesis of a seasonally
changing degree of PCP (see **Fig. 8**). Seasonal variations in the amount of PCP and its effect on $\delta^{13}$C has previously been recognized in monsoon regions (Johnson et al., 2006; Ridley et al., 2015). During In
P16, seasonal changes in incongruent dolomite dissolutionhost rock interaction or soil contribution
dominate the trace element variations of Mg versus Sr and Ba over PCP. However, since the main
source of carbon in Han-sur-Lesse cave waters is the vegetation cover above the cave (Genty et al.,
2001), IDD is not expected to change the $\delta^{13}C$ signal. For example, a case study carried out by Oster et
al. (2014) showed that an increase in IDD did not affect the $\delta^{13}C$ of the speleothem significantly, despite
a difference of ~0.5 ‰ in $\delta^{13}C$ between the limestone and dolomite component in the host rock. Since
$\delta^{13}C$ is not affected by IDD, the influence of PCP on the $\delta^{13}C$ remains observable. Increased degassing
related to PCP increases $\delta^{13}C$ in the summer season in all growth intervals. Indeed, similar as in P19P20,
for both P17 and P16 $\delta^{13}C$ minima occur within DCL, suggesting that these DCL layers in these intervals
were deposited by during seasonal periods of increased drip water discharge.

Observations from cave monitoring have shown that seasonal changes in cave temperature (11°C -
15°C) are the main driver of $\delta^{18}O$ variations in freshly deposited calcite (-7.0‰ - -6.2‰; Van
Rampelbergh et al., 2014). The $\delta^{18}O$ periodograms show that the seasonal $\delta^{18}O$ cycle is less developed
compared to $\delta^{13}C$ (**Fig. 4 6** and Supp. Mat. Fig. 3-5). This is also expressed in the annual stacks (**Fig. 7 9**).
For P19P20, there is tendency towards a positive correlation of $\delta^{13}C$ and $\delta^{18}O$ but in P17 and P16 this
is unclear. Although analysis analyses of recent calcite have clearly shown that $\delta^{18}O$ values are at least
partly controlled by the cave temperature, interpretation of the seasonal $\delta^{18}O$ changes is difficult due
to the reduced seasonal cyclicity in the $\delta^{18}O$ records compared to other proxies. However,
averagemean $\delta^{18}O$ values of speleothem calcite are obviously more depleted for P17 and P16
compared to P19P20 (**Fig. 3 4** and **Fig. 5**). Temperature changes over this period could in theory explain
some variation in $\delta^{18}O$, but only if these cooler temperatures resulted in cooler cave temperatures as
well, and only if fractionation was in equilibrium with ambient air, which is not certain for the 17th
century part of the Proserpine. In addition, this local temperature trend of ~1°C is not visible in the
long-term $\delta^{18}O$ record of **Fig. 3**, suggesting that other factors have a stronger influence on the oxygen
isotope fractionation in this speleothem (e.g. the $\delta^{18}O$ of precipitation). Therefore, the hypothesis put
forward here is that the lower averagemean $\delta^{18}O$ values of P16 point towards an increase in winter
precipitation above the cave, since Van Rampelbergh et al. (2014) has shown that winter precipitation,
such as the presence of snow, above Han-sur-Lesse cave causes a severe decrease in $\delta^{18}O$ of the
precipitation. Subsequently, this decrease is then transferred to the drip water and into the
speleothem calcite.

**5.3 Variability in the seasonal cycle**

The observed changes of the seasonal variations in $\delta^{13}$C, Mg, Sr and Ba between P20, P17 and P16

can only be explained by a change in the process controlling the seasonal variability in $\delta^{13}$C, Mg, Sr and Ba (**Fig. 9**). In the recent period, between 1960 and 2010 CE (P20), PCP is identified as the main driver for seasonal changes in Mg, Sr, Ba trace element concentrations. This hypothesis is supported by the $\delta^{13}$C variations. In the  16th century , Mg, Sr and Ba variations suggest that

IDD or soil activity, rather than PCP, dominate the seasonal signal.

Fairchild and Baker (2012) defined the term transfer function to describe the quantitative relation between  speleothem chemistry and changing cave environments or climate. This transfer function depends on several chemical and environmental variables, whose importance may change over time.

In this case, there is a (qualitative) change in  the transfer function  which causes PCP

to outweigh IDD and soil activity from the 16th century to modern times. This change in transfer function is likely  controlled by a change in the environment around the cave (e.g. change in land use or precipitation regime) since there are no indications for drastic changes in cave morphology over the last 500 years, as interpreted from the long term stable isotope ratio record (**Fig. 3**; Van Rampelbergh et al., 2015

). As mentioned above, a likely candidate for this environmental change is a change in land use (either natural or anthropogenic) between the 16th century and modern times, which could strongly influence soil activity, making soil activity a much less important factor in the trace element budget in the 17th and 20th century compared to the 16th century.

It is known that the strength of the acting transfer function can be used as a palaeoclimate proxy.

For example, Jamieson et al. (2016) demonstrated that the seasonal (anti-)correlation between $\delta^{13}$C

and U/Ca varies through time within a Common Era stalagmite from Belize. During drier years, reduced seasonal variability in prior aragonite precipitation causes U/Ca and $\delta^{13}$C to correlate more positively compared to wetter years. This illustrates how a transfer function can be regarded as a valuable palaeoclimate proxy. In any case, a certain external forcing is necessary for a  change  in transfer function to take place. A prerequisite for PCP to occur is the presence of sufficient karstic voids filled with a gas phase characterized by a lower pCO$_2$

than that with which the infiltrating waters previously equilibrated, or similarly lower pCO$_2$

concentrations in the cave that cause the formation of a large stalagmite above the speleothem (Fairchild and Treble, 2009). The presence of such karstic voids is dependent on the multi-annual to decadal recharge amount of the karstic aquifer. Indeed, the mean values of trace element concentrations imply  increased water availability during P16 and P17 compared to P20. More specifically, peaks in soil-derived trace element concentrations (Zn and Y) are higher for P16, pointing towards enhanced flushing and an increased seasonality in water availability as well as an increase in soil activity. This hypothesis is supported by the observation of marked increases in drip rates in the
cave during winter (Genty and Deflandre, 1998). An anthropogenic influence explains the higher
concentrations of Pb in P19P20, but may also partly explain the change that occurs between the 16th
century and later times. Sadly, no detailed information is available about the local changes in land use
in between the 16th and 17th centuries. However, it is inferred that forest cover above the Han-sur-
Lesse cave system was indeed reduced between the little ice age and modern times due to increased
anthropogenic activity (e.g. Van Rampelbergh et al., 2015). These changes mostly occurred on the
more humid slopes rather than the drier tops of the hills. Since the Salle du Dôme is situated under the
top of the hill (formed by the anticline structure mentioned in section **2.1**), changes in local forest cover
are relatively small, but cannot be neglected. In addition, trace element concentrations originating
from host rock dissolution (Mg, Sr, Ba and U) are significantly lower for P16, resulting from lower multi-
annual water residence time. Lastly, layers in P16 and P17 are up to three times thicker compared to
P19P20 (Fig. 2**Table 2, Fig. 3** and Supp. Mat. Fig. 6), which reflects higher growth rates. The positive
relationship between water supply and growth rate has been demonstrated in the past (Baker et al.,
1998; Genty and Quinif, 1996). In large and irregular shaped stalagmites, such as the Proserpine,
within-layer thickness can often be quite large (Baker et al., 2008). The long-term layer thickness
evolution shows a clear difference between the 17th century and present day. The significantly thinner
layers during recent times clearly indicate that less water is available compared to the 17th century.

A straightforward explanation for the observed wetter cave conditions during, in particular, P16 is an
increase in seasonal water excess. Gentry and Deflandre (1998) already demonstrated the strong
correlation between water access and drip rates in the Han-sur-Lesse cave system. An eElevated water
excess can be caused by an increase in precipitation or a decrease in temperature. A lLower
temperatures, especially during summer, results in a decreased evaporation of surface water.
Calculations of present-day potential evapotranspiration (PET) with the Thornthwaite equation
(Thornthwaite and Mather, 1957) for the period 1999-2012 show a negative water excess lasting from
May to September (**Fig. 82**). Although the Thornthwaite and Mather (1957) method does not include
vegetation effects, it is still a reliable tool to provide an idea of the effect of changes in the temperature
and/or precipitation on the PET (Black, 2007). The effect of a temperature decrease during summer
months on the water excess was simulated with an arbitrarily chosen 1°C temperature drop compared
to the 1999-2012 averagemean monthly temperature. Such a temperature drop appears to have only
a minor influence (**Fig. 82**). A hypothetical increase of total annual rainfall with 200 mm, equally spread
across 12 months, has a much larger effect on the water excess (**Fig. 82**). However, this would decrease
the length of the annual interval during which no recharge occurs (i.e. only during June-July instead of
May-September) providing less suitable conditions for dolomite dissolution to occur. It must be noted that rapid channelling of excess precipitation from intense rainfall events (especially during summer) into karst voids may decrease the relevance of seasonal changes in evapotranspiration (e.g. White and White, 2013). This further stresses the importance of changes in the precipitation regime on seasonality in water excess. The land use change proposed above would have influenced the infiltration regime. A change in forest cover since the 16th century, affecting soil thickness, would have the effect of increasing seasonality in soil-derived trace elements, especially in combination with an increase in precipitation seasonality. Therefore, two explanations could explain the seasonal-scale variability in trace element and stable isotope composition observed in the Proserpine speleothem. Firstly, a stronger seasonal distribution in the amount precipitation (with more winter precipitation) could have driven an increase in host rock interaction in the 16th century, whereas today very little seasonality in the amount of rainfall is observed. Secondly, a change in soil activity due to increased forestation in the 16th century compared to modern times could have produced the strong seasonality in soil derived trace elements in Proserpine due to seasonally enhanced leaching. The presented data do not allow conclusive distinction between these two processes, and a combination of both (one enhancing the other) is also possible.

**5.4 Implications for 17th century palaeoclimate**

The majority of Common Era continental palaeoclimate reconstructions are based on tree-ring data (D'Arrigo et al., 2006), although other records, for example historical documents (e.g., Dobrovolny et al., 2010), ice cores (e.g., Zennaro et al., 2014) or speleothems (e.g., Baker et al., 2011; Cui et al., 2012) are used as well. Over the last decades, consensus has been reached that changes in solar irradiance and volcanic activity are the main drivers of short-term (decadal to centennial) natural climate variability during the last millennium (e.g. Crowley, 2000; Bauer et al., 2003). Interpretations of the stable isotope and trace element proxies obtained on the Proserpine speleothem as well as the increased thickness of annual laminae show that a higher recharge state of the karstic aquifer characterizes the 17th century intervals compared to 1960-2010. Such an increase in recharge requires a decrease in evapotranspiration, which can result either from lower summer temperatures or higher total annual precipitation. Although it is difficult to discriminate between both, the effect of a total annual precipitation increase on the recharge is expected to be higher compared to a decrease in summer temperature (**Fig. 2**). As mentioned above, a change in land cover (both from anthropogenic or natural causes) could also have played a role by locally affecting the infiltration of this excess precipitation. Globally dispersed regional temperature reconstructions indicate that multi-decadal warm or cold intervals, such as the Medieval Warm Period or the Little Ice Age (LIA), are not global events. Yet, a global cooling trend starting at 1580 CE is observed in the majority of the reconstructions (PAGES 2k Consortium, 2013). Several palaeoclimate reconstructions agreed upon the occurrence of a cold period around 1600 CE, with negative temperature anomalies persisting in Europe at decadal and multi-decadal scales (Ljungqvist et al., 2012; Luterbacher et al., 2016; Masson-Delmotte et al., 2013). Reconstructions of European summer temperature provided by Luterbacher et al. (2016)

indicate that the coldest 11 and 51 year period since 755 CE in the area of Han-sur-Lesse cave occurred within the 17th century. These reconstructions showed a summer temperature decrease of 1 – 1.5°C

around 1600-1650 CE. Although the 17th century has been recognized as the coldest of the past twelve centuries, hydrological climate conditions appear close to the long-term mean (Ljungqvist et al., 2016), with no significant wetting or drying trend. However, to account for the differences between the 1960-

2010 interval and the 17th century observed in this study, an increase in the amount winter precipitation is needed, suggesting that climatic conditions were wetter during that time. Such a hypothesis is also supported by the depleted $\delta^{18}O$ values in P16, indicating an increase in winter precipitation.

**5.5 Implications for speleothem palaeoclimate studies**

This study complements a longer speleothem stable isotope compositional time series with shorter, higher resolution stable isotope and trace element records which capture changes on a sub-annual scale. One of the major advantages of this approach is the ability to study phase relationships between proxies at the seasonal level. The seasonal cycle is the strongest cycle in Earth's climate, and therefore allows relationships between speleothem proxy records to be tested in a context that is more familiar than that of decadal to millennial oscillations, which are less well understood. Comparing the seasonal expression of speleothem proxies also allows more straightforward comparison of proxy records with cave monitoring time series, which typically run on seasonal time scales (see Van Rampelbergh et al.,

2014). The high-resolution analyses carried out within the context of this study are relatively labour intensive. Measurements at this resolution are thus likely not feasible along the full (centuries- to millennia-long) growth period of a speleothem, even if seasonal variability is consistently recorded for the entire growth duration. Instead, we propose that the application of high-resolution, multi-proxy transects placed on strategic places along the growth axis of a speleothem (e.g. those parts where seasonal lamination is particularly well expressed) may be used to provide "snapshots" of seasonal proxy variability superimposed on longer term (decadal to millennial scale) variability in the record.

This exercise teaches us that the expression of trace element records should not be seen as a result of a constant transfer function, but rather as a complex interplay between chemical, speleological and environmental variables whose influence on the chemistry and (micro)morphology of the speleothem changes over time. Such changes in the transfer function have implications for the interpretation of the longer, lower resolution proxy records. This study therefore highlights the importance of including multiple proxies (e.g. trace element in combination with stable isotope, sedimentary and/or crystallographic analyses) as well as multiple sampling densities (high- and low resolution sampling) to reliably interpret speleothem archives in terms of climate and environmental evolution.

**6. Conclusions**

This study of annual trace element and stable isotope ($\delta^{13}$C and $\delta^{18}$O) variations over three different time intervals of the annually laminated Proserpine stalagmite from the Han-sur-Lesse Cave (Belgium)

shows that seasonal changes in Mg, Sr and Ba during the recent period (1960-2010) suggest a strong effect of prior calcite precipitation, caused by lower water availability during summer. In the 17$^{th}$

century (1600 CE ± 30 and 1640 CE ± 30), however, Mg is in antiphase with Sr and Ba. This implies that another process overwrites the PCP dominated seasonal cycle in these trace elements. A varying degree of incongruent dolomite dissolution, with more dissolution occurring during summer when water residence times in the epikarst are longer, or a more dominant influence of soil activity on the trace element budget due to changes in land use is the mostare plausible hypothesishypotheses, with more dissolution occurring during summer when water residence times in the epikarst are longer. The transfer function governing the trace elements concentrations in the Proserpine speleothem, is driven over the last centuries by changing contributions of PCP, or a varying degree of dolomite dissolution, and soil leachingdepends on water-rock interaction. Stable isotope ratios ($\delta^{13}$C and $\delta^{18}$O), soil derived trace element concentrations (Zn, Y and Pb) and speleothem morphology indicate that the multi- annual recharge of the epikarst was higher in the 17$^{th}$ century. The change in the response of Mg, Sr and Ba in the Proserpine speleothem to environmental changes was identified to be climate- drivendriven by climate or by changes in land use. and It likely may results from an recharge increase in recharge caused by a combination of lower summer temperatures and an increase in the amount of winter precipitation in the 17$^{th}$ century for the Han-sur-Lesse cave region, or by a change in the vegetation cover between the 16$^{th}$ and 17$^{th}$ century, which reduced the importance of soil processes on trace element compositions in the cave's drip waters. The effect of an increase in winter annual precipitation on the recharge is expected to be larger compared to a decrease in summer temperature.

The data obtained in this study clearly therefore showpoint towardss a stronger seasonal cycle in cave hydrology during the 17$^{th}$ century.

This high-resolution, multi proxy study provides a good example of how seasonal proxy transfer functionsthe relative importance of different processes on of trace element concentrations in speleothem calcite can change over time. Such anThis observation has implications for future speleothem-based paleopalaeoclimate reconstructions, since transfer functions for specific cave sites, determined by cave monitoring, are often assumed to remain constant when no drastic changes in the cave environment have occurred. As the change in trace element proxy transfer function observed in
this study is  driven by environmental change, this change by itself can serve as a valuable
palaeoclimate proxy.

**Author contributions**

Stef Vansteenberge and Sophie Verheyden designed the study. Stef Vansteenberge, Steven Goderis
and Stijn Van Malderen carried out LA-ICP-MS measurements. Stef Vansteenberge, Matthias Sinnesael
and Niels de Winter carried out stable isotope measurements. Stef Vansteenberge carried out the data
processing and plotting with contributions from Steven Goderis, Niels de Winter and Matthias
Sinnesael. Frank Vanhaecke and Philippe Claeys provided laboratory facilities and supported the
measurements. Stef Vansteenberge, Niels de Winter and Matthias Sinnesael prepared the manuscript
with contributions from all co-authors. Niels de Winter, Matthias Sinnesael and Sophie Verheyden
revised the manuscript in response to review comments.

**Acknowledgements**

The authors would like to thank dr. Robert Andrew Jamieson and two anonymous reviewers for their
thoughtful comments on the first version of this manuscript, as well as handling editor dr. James
Thornalley for moderating the review process. All authors thank the Domaine des Grottes de Han S.A.
for allowing us to sample the stalagmites and carry out other fieldwork. Special thanks for M. Van
Rampelbergh, whose PhD research formed the base of this study. S. Vansteenberge thanks J. Van
Opdenbosch and A. Ndirembako for their help collecting the stable isotope data and D. Verstraeten
for the lab assistance. This research was funded by the VUB Strategic Research Funding (S.
Vansteenberge), FWO Flanders (M. Sinnesael and S. Goderis), IWT Flanders (N. J. de Winter), Research
grant G017217N (S. J. M. Van Malderen and F. Vanhaecke) and the Hercules Foundation (upgrade of
the VUB Stable Isotope Laboratory).

|  |  |  |  |  |  |  |  |  |
|---|---|---|---|---|---|---|---|---|
|  |  |  |  |  |  |  |  |  |
|  |  |  |  |  |  |  |  |  |
|  |  |  |  |  |  |  |  |  |

**Table 2̶1**

| Proxy | P̶1̶9̶P20 | P17 | P16 |
|---|---|---|---|
| A̶v̶e̶r̶a̶g̶eMean Layer Thickness and growth rate | Thin: 0.382 mm
Larger variations (RSD = 28.9%)
U-Th mean growth rate: 0.564 mm/yr | Thick: 1.096 mm
Smaller variations (RSD = 6.3%)
U-Th mean growth rate: 1.34 mm/yr | Thick: 1.135 mm
Smaller variations (RSD = 9.5%)
U-Th mean growth rate: 0.910 mm/yr |
| $\delta^{18}O$ | Strong seasonality: tendency towards in phase correlation with $\delta^{13}C$
**Partially T-controlled, but other processes as well** | Weak to no seasonality: unclear relation with $\delta^{13}C$ | Weak to no seasonality: unclear relation with $\delta^{13}C$ |
| $\delta^{13}C$ | Clear $\delta^{13}C$ cycle:
Low $\delta^{13}C$ mostly in DCL but not always
**$\delta^{13}$C driven by seasonal changes in PCP** | Clear $\delta^{13}C$ cycle:
Low $\delta^{13}C$ always in DCL
**$\delta^{13}$C driven by seasonal changes in PCP** | Clear $\delta^{13}C$ cycle:
Low $\delta^{13}C$ always in DCL
**$\delta^{13}$C driven by seasonal changes in PCP** |
| Mg and Sr - Ba | Good in phase correlation
**Mg, Sr and Ba driven by seasonal changes in PCP** | Phase relation not clear
**Transition period between P16 and P̶1̶9̶P20 hydrological regimes** | Anti-phase correlation between Mg and Sr, Ba
**Seasonally occurring IDD dominates over PCP** |
| Zn, Y and Pb | Weak seasonality in Zn and Y,
Strong seasonality in Pb
**Decreased flushing, anthropogenic Pb enrichment** | Weak seasonality in Zn, Y and Pb
**Decreased flushing** | Very strong seasonality in Zn and Y, weak seasonality in Pb
**Enhanced flushing** |
| U | Strong seasonality, antiphase with Mg, Sr and Ba
**No PCP control, scavenging** | Weak seasonality antiphase with Sr and Ba
**scavenging** | No seasonality |
| Remarks | Link with trace elements and layering is challenging | Link with trace elements and layering is challenging | Link with trace elements and layering is challenging |

**Table 1: Schematic overview providing the observed changes and interpretation for the different proxies of P̶1̶9̶P20, P17 and P16. PCP = prior calcite**
**precipitation, IDD = incongruent dissolution of dolomite, DCL = dark compact layers, WPL = white porous layers**

| Isotope | $^{25}$Mg | $^{27}$Al | $^{29}$Si | $^{31}$P | $^{34}$S | $^{39}$K | $^{55}$Mn | $^{57}$Fe |
|---|---|---|---|---|---|---|---|---|
| LOD (µg g$^{-1}$) | 4.0 | 9.0 | 100 | 1.0 | 7.0 | 7.0 | 0.08 | 4.0 |
| Isotope | $^{66}$Zn | $^{85}$Rb | $^{88}$Sr | $^{89}$Y | $^{137}$Ba | $^{208}$Pb | $^{232}$Th | $^{238}$U |
| LOD (µg g$^{-1}$) | 0.2 | 0.03 | 0.08 | 0.01 | 0.1 | 0.008 | 0.0005 | 0.0001 |

**Table 2**: Overview of limits of detection (LOD) of trace elements measured for this study using LA-ICP-MS.

**FIGURE **

[Figure]

**(2 column figure)** **Figure 1:** (A) Location of the Han-sur-Lesse Cave system (50°06'51"N, 5°12'12"E) with the entrance and exit of the Lesse River, the Salle-du-dome and the Père-Noël Cave. North is upwards (B) Map showing the location of the Proserpine stalagmite within the Salle-du-Dome. The insert shows the position of the core retrieved from the speleothem. Images adapted from Van Rampelbergh et al. (2014, 2015). (C) Lithological column of the Devonian strata in the Synclinorium of Dinant including the two formations (Fromellennes and Mont d'Haurs) in which the Han-sur-Lesse cave is situated. Note the presence of partly dolomitized

[Figure]

**Figure 2**: Chart showing the calculated theoretical amount of water excess calculated with the Thornthwaite equation (Thornthwaite and Mather, 1957), based on temperature and precipitation data near Han-sur-Lesse cave from 1999 to 2012 (Royal Meteorological Institute, KMI). X-axis represents the months from January (1) to December (12).

[Figure]

**Figure 23:** A. Overview of long (~500 yr) record of stable isotope ratios and annual
layer thickness through the Proserpine speleothem measured by Van Rampelbergh et al. (2015). Red
boxes indicate the locations of high-resolution transects discussed in this study. B. The three studied
growth periods P20 (1960-2010 CE), P17 (1633-1644 ± 30 CE) and P16 (1593-1605 ± 30 CE). The
yellow rectangles mark the sections that were drilled/sampled for $\delta^{13}$C and $\delta^{18}$O analysis, the red lines
represent the LA-ICP-MS transects. Numbers in grey indicate the observed layer couplets. Note that
the images (**B**) of parts of the Proserpine speleothem that contain the three growth periods are
oriented vertically (top of the picture upwards), but that the transects themselves are oriented parallel
to the local growth direction (perpendicular to the growth laminae marked in blue). This results in transect P17's orientation at a ~25° angle with respect to the image (and the vertical).

[Figure]

**Figure 4:** Stable isotope ratios and trace element variations plotted against distance for P20, P17 and
P16. Blue bars mark the DCC laminae. The left side represents the youngest layers. All stable isotope
ratios are expressed as ‰ VPDB, while trace element concentrations are reported in ppm. Red bars
indicate years used for annual stack (**Fig. 7 and 9**).

[Figure]

(1.5 column figure) **Figure 3**5: Ranges of the stable isotope (left) and trace element data (right). For
the stable isotope ratios, the data mark the mean ( diamonds) and the standard
deviation (1σ) of the distribution. For the trace element concentrations, the boxes represent the
minimum and maximum values and the white diamonds mark the median. Numbers on top of the
bars represent the percentage of the data that is below the calculated detection limit.

[Figure]

(1 column figure) **Figure 46:** Periodograms (FFT) of δ¹³C, Mg, Zn and P measured in P16 to illustrate
how the quality of a proxy to record the seasonal cycle can be studied. The red line represents the 95%
confidence level. δ¹³C is taken as a reference. The periodograms include two examples of proxies with
a distinct peak in the seasonal frequency band of 0.8 mm⁻¹(Mg and Zn) and one proxy with no peak (P)
in the seasonal frequency band. Periodograms for all periods are provided in the supplementary
material.

(2 column figure) **Figure 5:** Stable isotope ratios and trace element variations plotted against distance
for P19, P17 and P16. Blue bars mark the DCC laminae. The left side represents the youngest layers.
All stable isotope ratios are expressed as ‰ VPDB, while trace element concentrations are reported in
ppm. Red bars indicate years used for annual stack (**Fig. 5 and 6**).

[Figure]

(1 column figure) **Figure 67:** Annual stacks of the trace element proxies. Y-axis: concentrations (μg g⁻
¹); x-axis: sub-annual increment (1 = January, 12 = December). Blue shaded areas indicate the winter
season. For the years used, see Fig. 45.

[Figure]

**Figure 8**: Cross plot showing the relationship between monthly average trace element (Mg, Sr, Ba) concentrations and $\delta^{13}C$ values in the three growth periods. Plots are labelled with the Pearson's r values indicating the strength and direction (phase relationship) of the correlation between the variables. Note the change in phase relationship between $\delta^{13}C$ and Sr and Ba, while $\delta^{13}C$ and Mg remain positively correlated.

(1.5 column figure) **Figure 79**: Annual stacks of δ¹³C (black) and δ¹⁸O (red). Dashed lines mark the 2σ
uncertainty. The x-axis represents one year. Blue shaded areas indicate the winter season. For the
years used, see Fig. 54.

(1 column figure) **Figure 8**: Chart showing the calculated theoretical amount of water excess calculated
with the Thornthwaite equation (Thornthwaite and Mather, 1957), based on temperature and
precipitation data near Han-sur-Lesse cave from 1999 to 2012 (Royal Meteorological Institute, KMI).
X-axis represents the months from January to December.

Commented [NdW1]: Move up to section 2.1 to describe modern climate